



# Mapping surface hoar from near-infrared texture in a laboratory

James Dillon[1], Christopher Donahue[2], Evan Schehrer[1], Karl Birkeland[3], Kevin Hammonds[1]

[1]Department of Civil Engineering, Montana State University, Bozeman, MT, USA
[2]Department of Geography, Earth, and Environmental Sciences, University of Northern British Columbia, Prince George, British Columbia, Canada
[3]USDA Forest Service National Avalanche Center, Bozeman, MT, USA

*Correspondence to*: James Dillon (james.dillon013@gmail.com)

**Abstract.** Surface hoar crystals are snow grains that form when water vapor deposits on the snow surface. Once buried, surface hoar creates a weak layer in the snowpack that can later cause large avalanches to occur. The formation and persistence of surface hoar are highly spatiotemporally variable making its detection difficult. Remote sensing technology capable of detecting the presence and spatial distribution of surface hoar would be beneficial for avalanche forecasting, however this capability has yet to be developed. Here, we hypothesize that near-infrared (NIR) texture, defined as the spatial variability of reflectance magnitude, may produce an optical signature unique to surface hoar due to the grains distinct shape and orientation. We tested this hypothesis by performing reflectance experiments in a controlled cold laboratory environment to evaluate the potential and accuracy of surface hoar mapping from NIR texture using a near-infrared hyperspectral imager (NIR-HSI) and a lidar operating at 1064 nm. We analyzed forty-one snow samples; three of which were surface hoar and 38 that consisted of other grain morphologies. When using NIR-HSI under direct and diffuse illumination, we found that surface hoar displayed higher NIR texture relative to all other grain shapes across numerous spectral bands and a wide range of spatial resolutions (0.5 - 50 mm). Due to the large number of spectral and spatial resolution combinations, we conducted a detailed samplewise case study at 1324 nm spectral and 10 mm spatial resolutions. The case study resulted in the median texture of surface hoar being 1.3 to 8.6 times greater than the 38 other samples under direct and diffuse illumination ($p <$ 0.05 in all cases). Using lidar, surface hoar also exhibited significantly increased NIR texture in 30 out of 38 samples, but only at select (5 – 25 mm) spatial resolutions. Leveraging these results, we propose a simple binary classification algorithm to map the extent of surface hoar on a pixelwise basis using both the NIR-HSI and lidar instruments. The NIR-HSI under direct and diffuse illumination performed best, with a median accuracy of 96.91% and 97.37%, respectively. Conversely, median classification accuracy with lidar was only 66.99%. Further, to assess the repeatability of our method and demonstrate mapping capacity, we ran the algorithm on a new sample with mixed microstructures, with accuracy of 99.61% and 96.15% for direct and diffuse, respectively. As NIR-HSI detectors become increasingly available, our findings demonstrate the potential of a new tool for avalanche forecasters to remotely assess the spatiotemporal variability of surface hoar, which would improve avalanche forecasts and potentially save lives.



## 1 Introduction and background

Mountainous and polar snowpacks are commonly blanketed by surface hoar, unique ice crystals that grow when water vapor deposits on the snow surface (Horton and Jamieson, 2017). Hoar crystals can grow to several centimeters in length and stand in a predominately vertical (but often quite variable) orientation atop the snow surface. Surface hoar is a prominent concern for avalanche forecasters due to its propensity for creating weak layers in the snowpack that can cause large avalanches. Generally, a weak layer is formed due to a snow layer being weakly bonded to the slab above, as in the case of an ice lens, or being of low shear strength, such as with surface hoar. Once buried, surface hoar layers are prone to fracture propagation and avalanche release (Horton and Jamieson, 2017; Jamieson and Schweizer, 2000). For instance, Birkeland (1998) found that nearly one-third of large natural avalanches in southwestern Montana failed on buried layers of surface hoar, while Jamieson and Schweizer (2000) similarly observed in a Swiss dataset that 40% of avalanches released on a surface hoar weak layer.

Surface hoar formation and persistence is highly spatiotemporally variable. Its presence is difficult to predict because it depends on the complex spatial distributions of atmospheric water vapor, precipitation, wind, radiation, and vegetation that are present in polar and mountainous environments. Temporally, surface hoar can be promptly destroyed by environmental influences, such as wind, or it can persist for weeks (Champollion et al., 2013; Lutz and Birkeland, 2011). The formation of surface hoar can occur across the landscape or in isolated sub-slope regions. Therefore, before becoming a buried weak layer in the snowpack, surface hoar detection is an ideal application for remote sensing.

Optical remote sensing retrievals of snow and ice commonly use near-infrared (NIR) wavelengths where ice is absorptive, and reflectance is sensitive to microstructure. In the NIR, snow grain size is the primary driver of snow reflectance and albedo; the increased path length of light through ice yields a reduction in reflectance. Despite the complex shapes of snow grains, representing snow as a collection of ice spheres with radius ($r_e$), known as the optical or effective snow grain size, is commonly used for simulating snow reflectance and albedo (Grenfell and Warren, 1999). The effective grain size is related to the physical snow microstructure through the ice surface area-to-volume ratio (or specific surface area; SSA) by the following relationship:

$$r_e = \frac{3}{SSA} \tag{1}$$

In practice, the non-linear inverse relationship between NIR reflectance and $r_e$ is leveraged to map $r_e$ from reflectance measurements (e.g., Nolin and Dozier, 2000). Because grain size controls broadband NIR albedo, estimates of $r_e$ have historically dominated physical snow surface characterization, constituting a major goal of snow optics.

In recent decades, the use of NIR hyperspectral imaging (NIR-HSI), and (to a lesser extent) light detection and ranging (lidar), have enhanced $r_e$ mapping efforts. Hyperspectral instruments have superior spectral resolution relative to broadband or multispectral reflectance measurements, producing a nearly continuous measured spectra for each pixel in an image. Lidar, on the other hand, holds key advantages as an active remote sensor. Lidar units scan a surface by emitting rapid pulses of light (most commonly at a NIR wavelength) and record both the relative strength of backscattered light after reflecting off a target, and the two-way travel time. These two measurement types can be used independently; the travel time



is used to measure surface topography, whereas the backscattered magnitude has been demonstrated as a useful measure of optical properties. Although far less validated than traditional passive reflectance measurements, lidar backscatter may provide adequate $r_e$ estimates (Yang et al., 2017).

Despite advances in snow optics, Horton and Jamieson (2017) note the fundamental disconnect between snow surface characterizations conducted by the remote sensing community (i.e., $r_e$ mapping), and the physical properties relevant

to avalanche release. For avalanche forecasters, characterizing snow surface microstructure with the morphological grain shapes defined in the International Classification for Seasonal Snow on the Ground (ICSSG, Fierz et al., 2009) and their associated mechanical properties is critically important because these mechanical properties help determine how well new snow will bond to the old snow surface. Hence, a forecaster would much prefer a map identifying a potential future weak layer, like surface hoar, than a map of $r_e$. Relating morphological grain shape to $r_e$ is difficult because effective grain size

only considers the path length of ice which is a complex function of grain shape, traditional grain size, bulk density, and other physical characteristics. Therefore, $r_e$ is not particularly useful for avalanche forecasting operations, although room for this adaptation does exist. While certain magnitudes of $r_e$ are generally related to grain shape (Domine et al., 2007; Matzl and Schneebeli, 2010), few studies have formally attempted to use NIR reflectance for mapping morphological grain shape instead of $r_e$. Further, the studies that have attempted this (e.g., Bühler et al., 2014; Horton and Jamieson, 2017) have found

that surface hoar crystals produce moderate reflectance signatures relative to other grain shapes, making them difficult to delineate from less-concerning snow microstructures based on NIR reflectance magnitude.

To the best of our knowledge, the only study to date that successfully identified surface hoar formation from NIR remote sensing was conducted by Champollion et al. (2013) in Antarctica. As opposed to evaluating magnitudes of NIR reflectance, the researchers leveraged a NIR texture signature, defined as the localized spatial variability in reflectance, to

classify the presence of surface hoar using an infrared camera and an 850 nm artificial light source. Simply put, the researchers found that a large, localized variance in NIR reflectance, as quantified by a contrast index, was strongly correlated with surface hoar crystals. Similarly, in the preliminary findings presented by Walter et al. (2023), the researchers quantified a 600% increase in NIR reflectance spatial variability during surface hoar formation, measured with a 905 nm lidar unit.

Here, we postulate that the physical phenomena for these findings include increasing specular contributions with surface hoar growth (Walter et al., 2023), as well as variable ice absorption and path length (Fig. 1b). Depending on how an incoming photon interacts with the relatively large, often vertically oriented surface hoar crystals, the photon could experience a wide range of ice path lengths and thus absorption. Further, it is thought that surface hoar plates promote specular reflectance, which can cause large portions of radiation to either return to the sensor or forward scatter toward the

snowpack and into the "radiation trap" of hoar crystals. As a result, *we hypothesize that the presence of surface hoar will coincide with a quantifiable increase in localized reflectance variance,* a texture signature, which could be used to map its distribution. If this is the case, it would enable fine spatial and temporal resolution mapping of surface hoar extent (prior to burial) in challenging to observe environments, particularly as NIR remote sensors and uncrewed aerial vehicles (UAVs)





become more cost-effective as forecasting tools. However, such a texture analysis has never been fully evaluated, rigorously

quantified for accuracy, or compared to a wide variety of well-defined microstructures to ensure that the NIR texture is

indeed a unique defining feature of surface hoar.

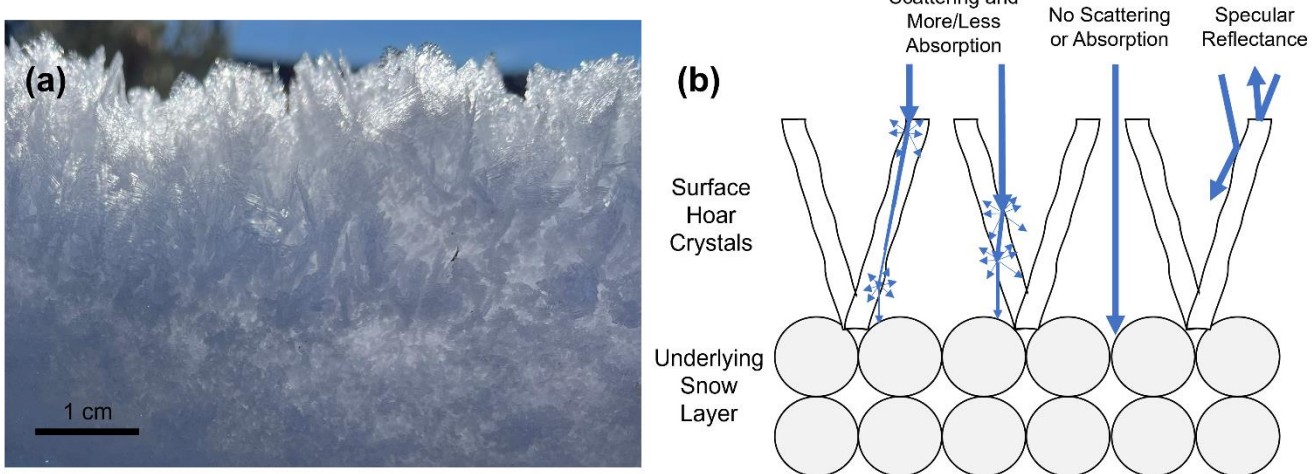

**Figure 1: Profile view of surface hoar atop an underlying snow layer (a) juxtaposed with an idealized schematic of light scattering when encountering a surface hoar layer (b). Because surface hoar crystals are typically large and vertically oriented, incoming**
**photons may experience a very long path length and thus substantial absorption before reaching the underlying snow layer (leftmost case). Depending on the angle of interaction, that path may be considerably shortened (middle left case), or, as these crystals tend to be modestly spaced, photons may evade the surface hoar crystals entirely and pass straight into the underlying snow layer (middle case). Further, specular contributions are thought to increase in surface hoar layers (rightmost cases), which can produce particularly high or low reflectance depending on the angle of interaction. Last, because the surface hoar crystals will**
**almost certainly have a different SSA than the underlying layer, light scattering back to the sensor directly from the surface hoar layer should further enhance the spatial variability of reflectance.**

To address this knowledge gap, we performed controlled cold laboratory experiments to determine whether NIR

texture can delineate the extent of surface hoar. We first created snow samples with varying grain shapes and physical

properties and quantified their microstructures using X-ray computed microtomography (micro-CT). Using both a compact

NIR-HSI and a terrestrial lidar unit, we scanned each sample under a variety of illumination conditions. We subsequently

analyzed the resulting maps of reflectance to produce measurements of NIR texture. Finally, we assessed the statistical

significance of increased texture in surface hoar samples. We used our results to inform optimal thresholds for classifying

surface hoar on a per-pixel basis, before analyzing the accuracy of our resulting classified data products.

## 2 Methodology

We aimed to prepare laboratory snow samples with a wide variety of well-defined grain shapes and microstructures, acquire

optical measurements, and perform a texture analysis towards delineation of surface hoar from other snow surface grain

shapes. Section 2.1 describes snow sample preparation and physical characterization, Sect. 2.2 describes the acquisition of



NIR-HSI and lidar data, Sect. 2.3 covers image texture analysis, Sect. 2.4 describes classification, and Sect. 2.5 evaluates the repeatability of our work.

## 2.1 Sample preparation and physical characterization

### 2.1.1 Sample preparation

We utilized Montana State University's Subzero Research Laboratory (SRL), a controlled cold laboratory environment, for sample preparation, testing and assessment. The snow used in these experiments was a combination of laboratory-grown crystals produced in the SRL's snowmaking apparatus, which is similar to the systems presented in Schleef et al. (2014) and Abe and Kosugi (2019), and natural undisturbed snow that we collected from the surrounding area. To ensure the snow was completely dry, we kept all samples in a cold room at -30° C and allowed them to equilibrate for at least 24 hours prior to evaluation. We prepared forty-one snow samples from twelve batches of differing snow grains (Fig. 2). From the bulk batches, we sieved snow grains through differing mesh sizes to further promote disparate microstructures (Table 1). The exception to this was surface hoar, which was grown in the laboratory atop rounded grains following the methods used by Stanton et al. (2016). Samples 25 and 26 consisted of in situ surface hoar growth at differing stages. Meanwhile, Sample 24 featured these surface hoar grains redistributed through a large (6.30 mm) sieve, in an effort to further examine the optical behavior of these grains after disrupting their typical vertical structure. Sample grain shapes included precipitation particles (PP), decomposing and fragmented precipitation particles (DF), rounded grains (RG), melt forms (MF), faceted crystals (FC), depth hoar (DH), and surface hoar (SH) (Fierz et al., 2009). We prepared snow samples to be microstructurally homogeneous, both laterally across the sample and vertically over sample depth, in a rectangular 38 cm x 23 cm sample holder. Sample thickness was 3.8 cm, over twice the largest optically active depth estimates for our sample microstructures in the NIR spectrum (Nolin and Dozier, 2000).





**Figure 2: Microscopy images of grains from each initial batch (left columns) and binary micro-CT cross-sections from representative samples (right columns). In the microscopy images, the grid size on the underlying blue grain card is 2 mm.**



**Table 1: Physical snow sample characteristics organized by primary grain shape and listed in order of decreasing surface area-to-volume ratio therein.**

| Sample # | Batch ID | Primary Grain Shape | Secondary Grain Shape(s) | Micro-CT SSA (mm$^{-1}$) | Micro-CT ρ (kg m$^{-3}$) | Sieve Size (mm) Passed | Sieve Size (mm) Caught | Notes |
|---|---|---|---|---|---|---|---|---|
| 1 | A | PP | PPrm, DF | 32.87 | 176 | 2.38 | 1.18 | |
| 2 | A | PP | PPrm, DF | 28.98 | 217 | 2.38 | - | |
| 3 | A | PP | PPrm, DF | 26.31 | 211 | 1.18 | 0.42 | |
| 4 | B | PP | PPgp, DF | 31.79 | 160 | 2.38 | 1.18 | |
| 5 | C | PP | DF | 33.10 | 94 | - | - | In situ fresh PP |
| 6 | C | PP | DF | 20.54 | 286 | 2.38 | 1.18 | |
| 7 | C | PP | DF | 20.45 | 280 | 0.85 | 0.42 | |
| 8 | C | PP | DF | 20.12 | 275 | 2.38 | - | |
| 9 | C | PP | DF | 18.39 | 303 | 1.18 | 0.85 | |
| 10 | D | DF | RG | 27.44 | 293 | 2.38 | 1.18 | |
| 11 | D | DF | RG | 25.85 | 323 | 0.85 | 0.42 | |
| 12 | D | DF | RG | 25.11 | 351 | 1.18 | 0.85 | |
| 13 | D | DF | RG | 20.77 | 365 | 2.38 | - | |
| 14 | E | DF | DFbk, RGwp | 16.27 | 374 | 0.85 | - | |
| 15 | F | DF | PP | 14.39 | 322 | 2.38 | - | |
| 16 | F | DF | PP | 13.79 | 312 | 2.38 | 1.18 | |
| 17 | F | DF | PP | 13.65 | 309 | 1.18 | 0.85 | |
| 18 | F | DF | PP | 12.99 | 382 | 0.85 | - | |
| 19 | G | FC | DH | 14.67 | 407 | 1.18 | 0.42 | |
| 20 | G | FC | DH | 11.32 | 448 | 2.38 | 1.18 | |
| 21 | G | FC | DH | 10.26 | 417 | 6.30 | 3.35 | |
| 22 | G | FC | DH | 10.05 | 472 | 6.30 | - | |
| 23 | G | FC | DH | 9.87 | 404 | 3.35 | 2.38 | |
| 24 | H | SH | RG | 14.52 | 213 | 6.30 | - | Re-sieved SH grains |
| 25 | H | SH | RG | 10.82 | 65 | - | - | In situ SH atop RGs |
| 26 | H | SH | RG | 7.50 | 94 | - | - | Smaller than S25 |
| 27 | I | RG | DF | 13.53 | 381 | 2.38 | 1.18 | |
| 28 | I | RG | DF | 13.08 | 419 | 1.18 | 0.85 | |
| 29 | I | RG | DF | 12.77 | 431 | 2.38 | - | |
| 30 | I | RG | DF | 12.43 | 489 | 0.85 | 0.42 | |
| 31 | I | RG | DF | 12.40 | 452 | - | - | S29 melt/refreeze |
| 32 | J | RG | DF | 13.76 | 394 | 0.85 | - | |
| 33 | J | RG | DF | 13.45 | 355 | 0.42 | - | |
| 34 | J | RG | DF | 10.66 | 460 | 1.18 | - | |
| 35 | K | RG | DF | 11.13 | 404 | 1.18 | 0.85 | |
| 36 | K | RG | DF | 10.84 | 428 | 0.85 | 0.42 | |
| 37 | L | MF | RG | 4.96 | 582 | 2.38 | 0.42 | |
| 38 | L | MF | RG | 3.69 | 545 | 6.30 | - | |
| 39 | L | MF | RG | 3.13 | 512 | 3.15 | 2.38 | |
| 40 | L | MF | RG | 2.88 | 467 | - | - | Refrozen in-situ |
| 41 | L | MF | RG | 2.37 | 433 | 6.30 | 3.15 | |

### 2.1.2 Physical characterization

We thoroughly characterized the physical properties of each sample, as summarized in Table 1. First, we performed
microscopy on representative grains from each batch prior to sieving (Fig. 2), and classified grain shapes using a crystal card



and lens following Fierz et al. (2009). After sieving and sample preparation, we collected micro-CT data from each sample using a Bruker SkyScan 1173 housed in a -10° C chamber within the SRL, generally following the protocol outlined by Donahue et al. (2021). To prepare samples, we used a cylindrical holder with 3 cm diameter x 4 cm length, which allowed for a voxel size of 14.5 μm. We obtained measurements using a 42 kV, 190 mA X-ray beam, 100 ms exposure time, with

each sample rotated 180° at 0.7° increments. After scanning, we performed thresholding of grey-scale images into ice and air phases by visual inspection (e.g., Fig. 2), and used a despeckling filter to remove white and black speckles. Reconstructions via the marching cubes method (Lorensen and Cline, 1987) allowed us to determine the volume and surface area in 3D, which we used to calculate the SSA and density of each sample.

## 2.2 Optical measurements

We examined NIR texture under a variety of illumination and viewing conditions, using both lidar and NIR-HSI independently. For NIR-HSI, we constructed laboratory setups for both direct (Sect. 2.2.1.1) and diffuse (Sect. 2.2.1.2) illumination conditions. As lidar (Sect. 2.2.2) produces its own direct illumination via laser irradiance, this was the only possible illumination condition for lidar analysis. Furthermore, it is well-understood that incidence angle impacts snow reflectance in the NIR spectral region; snow is nearly Lambertian under nadir incidence and predominantly forward

scattering at off-nadir incidence. To consider this, for each of the aforementioned illumination configurations/instruments, we collected data with: 1) the sample container perpendicular to the detector (nadir), and 2) tilted 10° away from the detector. Hereafter, these incidence angles of 0° and 10° are termed Ɵ. The result was six datasets representing each illumination/instrument/incidence combination for every sample. We acquired all optical data immediately prior to micro-CT analysis at a constant temperature of -10° C.

## 2.2.1 Hyperspectral imaging

We used a Resonon Inc. Pika NIR-640 near-infrared hyperspectral imager to map snow reflectance in the NIR (www.resonon.com). Donahue et al. (2021) provide a detailed description of the instrument. Briefly, the imager's spectral resolution ranges from 2.39 to 2.50 nm, and measures 336 bands across the NIR region from 891–1711 nm. It constructs a 2D image containing the full spectrum in each pixel by collecting the image line by line, known commonly as a "push

broom" or "line" scanner. Thus, to collect an image, the camera must move (translating or rotating) relative to the scene, or the scene must move relative to the imager. We used a Resonon benchtop linear scanning stage to move the sample beneath the sensor.

### 2.2.1.1 Direct illumination data collection

We positioned the hyperspectral imager above the linear translating stage that held the samples. For more details on the

benchtop apparatus, see Donahue et al. (2022). The lens of the imager is surrounded by a set of four halogen lamps that produce direct illumination (Fig. 3a). The halogen lamps and lens of the imager are at a height of 38 and 47 cm above the





snow surface, respectively. We used a large Spectralon white diffuse reflectance panel to perform a pixel-by-pixel calibration, resulting in a reflectance factor (R) measured for each band in every individual pixel of the image.

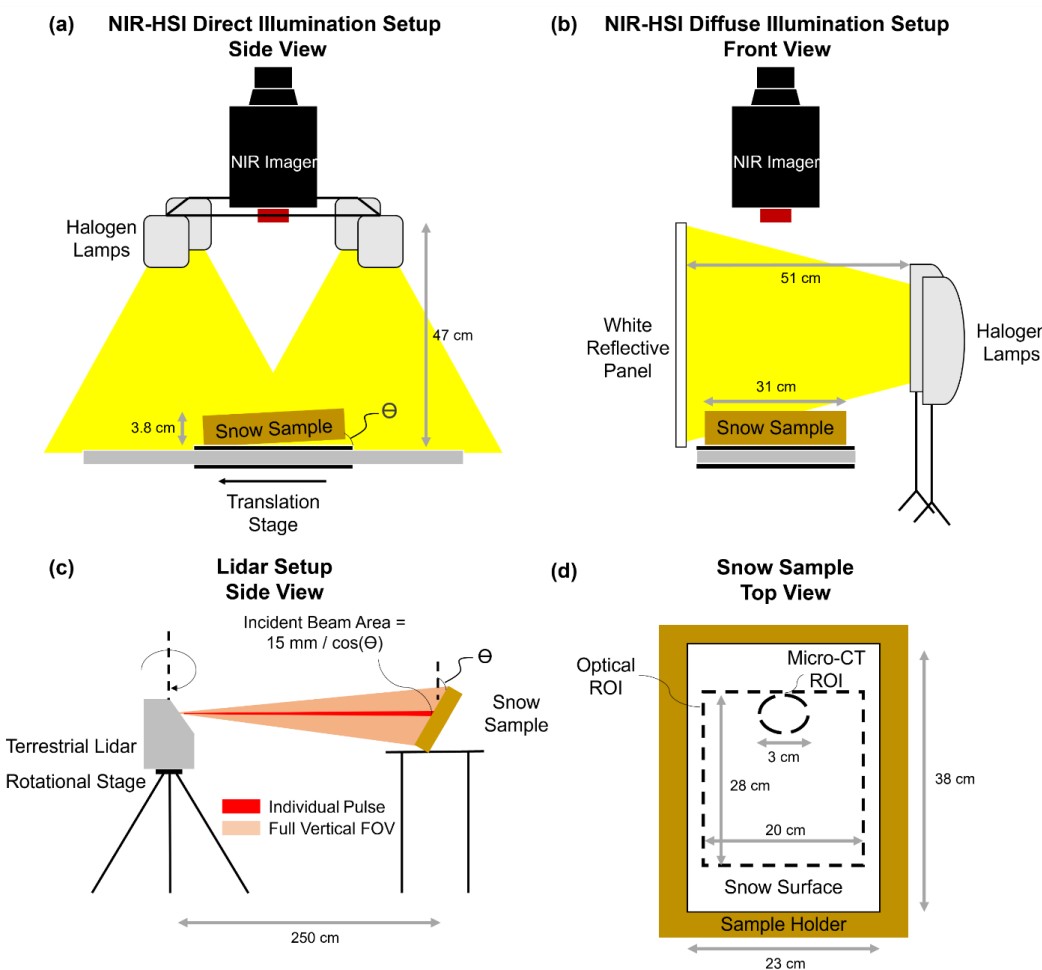

**Figure 3: Laboratory data collection schematic for hyperspectral imaging under direct (a) and diffuse (b) illumination conditions, as well as for lidar (c). Data regions-of-interest (ROIs) within the snow sample are illustrated in (d). Angle Ө refers to the viewing incidence angle (as well as the illumination incidence angle, in the case of both lidar and NIR-HSI under direct illumination).**

### 2.2.1.2 Diffuse illumination data collection

For the diffuse illumination setup, we kept the imager mounted in the same orientation, but removed the set of four halogen lights used for direct illumination. Instead, we positioned two larger softbox diffuse halogen lamps (Westcott uLite) perpendicular to the snow sample surface (Fig. 3b). Aiming the lights at a large white panel on the opposite side of the snow sample, 51 cm away mimicked diffuse hemispheric illumination conditions with no direct component. Once again, we used a Spectralon panel to convert raw data to reflectance images.





### 2.2.1.3 Hyperspectral data processing

Initial processing took place in Resonon's proprietary Spectronon software, and analyses thereafter performed in Rstudio. We began by truncating the reflectance data from each dataset to a central region-of-interest (ROI) encapsulating the micro-CT ROI (Fig. 3d). Our goal was to exclude edge effects from mixed pixels/points, particularly considering lidar beam dilation when samples were tilted off-nadir. Resulting NIR-HSI ROIs contained 224,000 pixels with a spatial resolution of 0.5 mm. Reflectance images were produced from 188 of the 336 available bands, ranging from 951 – 1403 nm, trimmed to
reduce noise at the lower end of the imager spectral range and to exclude regions where snow and ice are scarcely reflective. Example snow sample reflectance maps from a single band centered at 1030 nm, the location of a prominent ice absorption feature, (hereafter $R_{1030}$) are illustrated in Fig. 4c and 4d. The last factor we sought to examine was spatial resolution, considering that if a NIR texture signature specific to surface hoar does exist, then it is likely resolution-dependent. Thus, we coarsened all datasets to spatial resolutions of 1, 2.5, 5, 10, 25, and 50 mm (hence two orders of magnitude), as an attempt to
mimic the finer spatial resolutions achievable by UAV-mounted systems.

### 2.2.2 Lidar

### 2.2.2.1 Data collection

We used a Riegl VZ-6000 terrestrial laser scanner mounted to a tripod. The scanner achieves vertical (line) scanning via an oscillating mirror while moving horizontally on a rotating head (Fig. 3c). The maximum vertical scan field-of-view (sFOV)
is 60° - 120° from zenith and selectable therein, while the horizontal sFOV can range from 0° to a full 360° panorama. The laser operates in the NIR range narrowly around a central wavelength of 1064 nm. We set vertical and horizontal angular increments to 0.01° to maximize resolution (point density) and selected a pulse repetition frequency of 300 kHz. The initial laser beam diameter upon exiting the scanner is 15 mm with a divergence of 0.12 mrad. We positioned the scanner about 2.5 m from the snow samples. The duration of each scan was approximately 1 minute. The resulting data product for lidar is a
"cloud" of discrete vector data points, called returns. Similar to the hyperspectral imaging setup, we used a Lambertian reflectance standard to convert return power to bidirectional reflectance factor for each individual point in the clouds.

### 2.2.2.2 Lidar data processing

Initial processing of point clouds took place in Riegl's RiScan application, and analyses thereafter in Rstudio. As with NIR-HSI data, we trimmed point clouds to a central ROI (Fig. 3d). After truncation, the average point cloud contained 80,000
returns, with spacing of ~ 1.4 pts/mm$^2$. In order to perform pixelwise operations and to better compare with hyperspectral imagery, we converted the point cloud into an image of continuous pixels. Using a Delaunay triangulation, we interpolated point clouds into images with 1 mm spatial resolution, resulting in 56,000 pixels per dataset. The Riegl VZ-6000 lidar unit operates narrowly around a central wavelength of 1064 nm. Therefore, unlike NIR-HSI, the lidar is only capable of measuring bidirectional reflectance specifically at 1064 nm (hereafter $R_{1064}$; e.g., Fig. 4e and 4f). This wavelength occurs on



the shoulder of the ice absorption feature centered at 1030 nm (Inset of Fig. 4e). Last, as with NIR-HSI, we coarsened all

lidar datasets to the resolutions listed in Sect. 2.2.1.3 to examine the influence of spatial resolution on NIR texture.

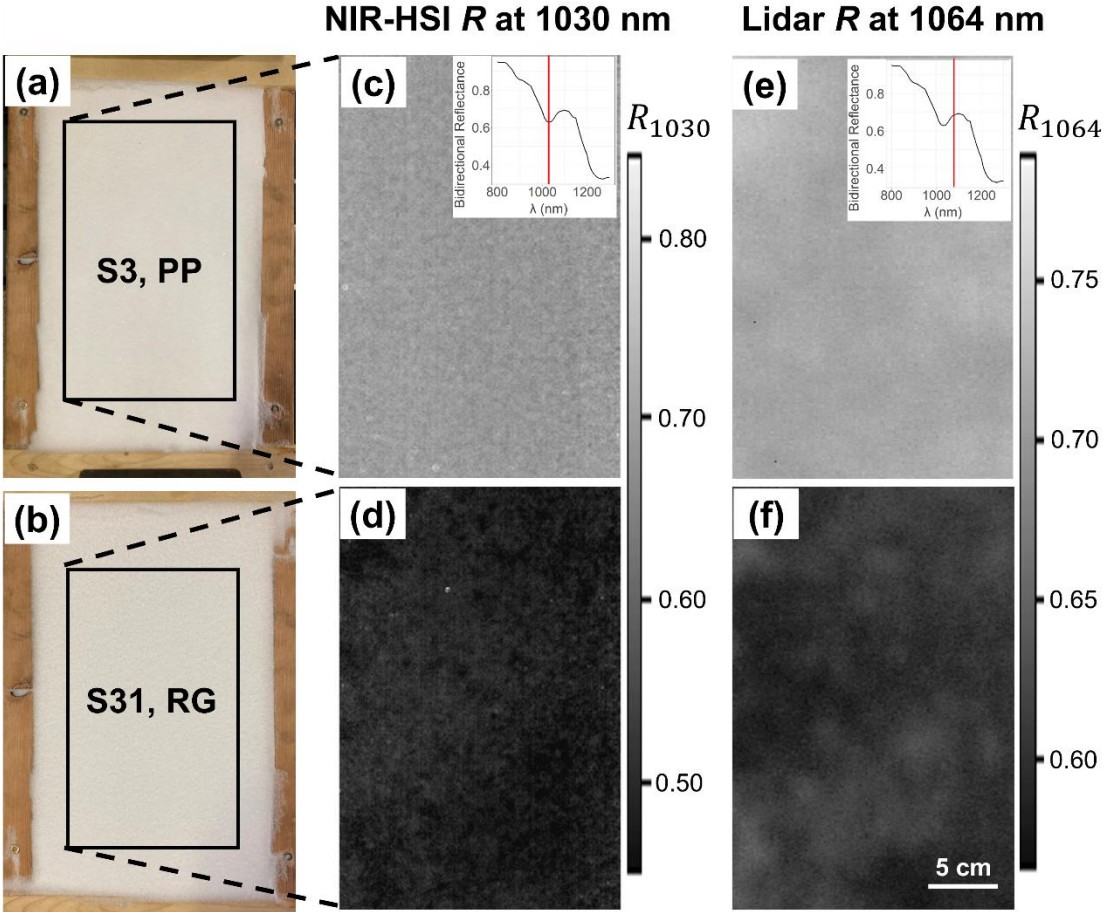

**Figure 4: Visible photographs of two different snow samples (a and b) are juxtaposed with greyscale images of hyperspectral reflectance at 1030 nm (c and d), as well as lidar-derived reflectance at 1064 nm (e and f). Samples are shown under direct**
**illumination at Θ = 0° and at their original resolutions, prior to any coarsening. The data in the top row depicts Sample 3, a snow microstructure with relatively high specific surface area (26.31 mm$^{-1}$) compared to Sample 31 in the bottom row, which had a specific surface area of 12.40 mm$^{-1}$. The insensitivity of reflectance to snow microstructure in the visible range, as opposed to the dramatically different reflected magnitudes in the NIR, is apparent. The inset figures illustrate the location of the bidirectional reflectance measurements relative to the ice absorption spectra. These example spectra were produced using the Asymptotic**
**Radiative Transfer model (Kokhanovsky and Zege, 2004).**

## 2.3 Texture analysis

### 2.3.1 Moving window focal analysis

To restate our hypothesis, we anticipated that surface hoar would display heightened NIR variability, or texture, relative to

other snow surface grain shapes due to the physical phenomenon illustrated in Fig. 1. Therefore, we sought to evaluate

localized variability of reflectance. To achieve this, we performed a moving window focal analysis to create maps of local



standard deviation. Using $R_{1030}$ from NIR-HSI, a demonstration of both coarsening and subsequent moving window analysis is presented in Fig. 5 for Sample 20. Beginning with a map of R at either the native 0.5 mm resolution (Fig. 5a), or coarsened resolutions of 5 mm and 50 mm (Fig. 5b – 5c, respectively), a 9-pixel neighborhood is placed around a central pixel. The standard deviation of $R_{1030}$ (hereafter $\sigma_{1030}$) is calculated within the window via Equation 2:


$$\sigma = \sqrt{\frac{\sum_{i=1}^{3}\sum_{j=1}^{3}(R_{ij} - \bar{R})^2}{N}}$$
(2)

where $R_{ij}$ is the reflectance value at row i and column j in the 3x3 grid, $\bar{R}$, is the mean of the reflectance values in the 3x3 grid, and N is the total number of pixel values (nine in this case). The resulting value of $\sigma$ is assigned to the central pixel.

We handled edges by truncating the window size when necessary, such that corner pixels only considered three neighboring cells, for example. Moving the window across each $R_{1030}$ map and evaluating every pixel independently yields a map of $\sigma_{1030}$ (Fig. 5d – 5f). Hence, these maps depict NIR reflectance texture, rather than magnitude. We underwent this process for all datasets, thus each of the 188 NIR-HSI bands and lidar-derived $R_{1064}$, for each illumination condition and incidence angle.

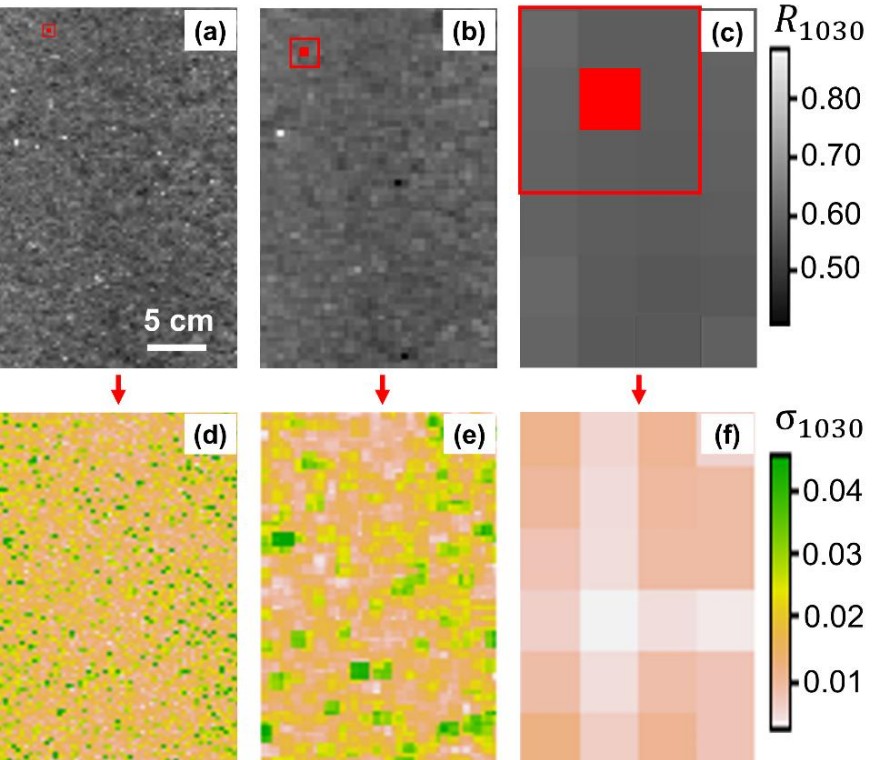

**Figure 5: A map of $R_{1030}$ from NIR-HSI at its raw resolution of 0.5 mm (a) is coarsened by two orders of magnitude to 5 mm (b) and 50 mm (c). For each resolution, the localized standard deviation, $\sigma_{1030}$, is calculated using a nine-pixel moving window analysis (d – f). The data shown is from Sample 26 under direct illumination at an incidence angle of $\Theta = 0°$.**



### 2.3.2 Spatial and spectral texture analysis

Once texture maps were derived for each dataset, we determined which spatial resolutions were best for surface hoar
delineation, assuming some degree of resolution-dependence. Furthermore, leveraging the spectral data provided by NIR-
HSI, we examined the relationship between NIR texture and wavelength. We began by grouping the pixelwise σ
distributions from surface hoar samples 24 – 26 for each dataset. Similarly, we grouped σ distributions of all other samples
(hereafter termed "Other", meaning a microstructure other than SH). Next, we calculated median values of both grouped
distributions for each instrument, band, and spatial resolution, and determined the percent difference in medians for each
case. Using heat maps and line graphs, we evaluated σ distributions and resulting differences in medians (i.e., Δ M(σ)). This
allowed us to determine the spatial resolutions and, in the case of NIR-HSI, the wavelengths, that maximized the difference
between SH and Other microstructure medians. We note that a larger difference in texture medians between SH and Other
samples should allow for greater ease of SH classification.

### 2.3.3 Samplewise analysis and significance testing

We next examined how texture varied between individual samples, rather than considering SH samples against an
aggregation of other microstructures (Sect. 2.3.2). As a case study, we selected a single band and spatial resolution for each
instrument. For NIR-HSI, we chose the band centered at 1324 nm. The results of our spatial and spectral analysis (described
in Sect. 2.3.2; results to be discussed in Sect. 3.1) indicated that this was an optimal wavelength under both direct and diffuse
illumination. The lidar unit has only one band, centered at 1064 nm. We elected to optimize spatial resolutions based on the
results in Sect. 3.1 as well, and thus we selected 10 mm for NIR-HSI and 5 mm for lidar. To determine if surface hoar
textures are significantly larger than those of differing microstructures, we compared distributions and median values of σ
across samples. As in Sect. 2.3.2, we grouped the pixelwise σ distributions of surface hoar (Samples 24 – 26) for each
dataset. We then compared the median value of this grouped SH distribution against samplewise medians. Specifically, we
performed one-sided, one-sample t-tests to assess the grouped median against the median σ of each sample. The flow chart in
Fig. 6 describes the processing workflow from raw reflectance images through statistical analyses for the case study. Our
hypotheses can be summarized as follows, where X describes an individual sample number:

$$H_0:\ M(\sigma^{S24:S26})\ =\ M(\sigma^{S(X)})$$
$$H_A:\ M(\sigma^{S24:S26})\ >\ M(\sigma^{S(X)});\ \ \alpha = 0.05$$




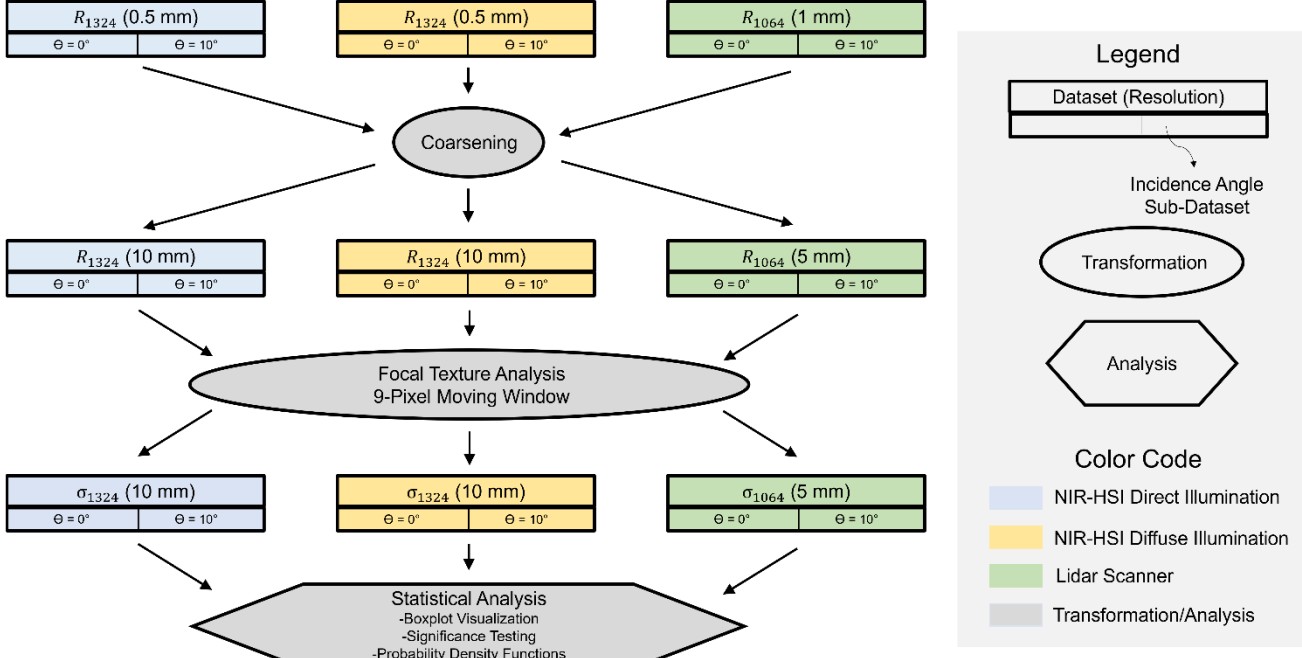

**Figure 6: Flowchart of samplewise data processing and statistical analysis workflow.**

## 2.4 Classification algorithm

Continuing our samplewise case study (Sect. 2.3.3), we next produced classified maps of surface hoar. We determined

optimal threshold values of σ to delineate surface hoar from other snow surface shapes on a per-pixel basis. In addition to visualizing distributions (across all pixels) of σ with boxplots for each sample, we also constructed probability density functions (PDFs). As in Sect. 2.3.2, we grouped the distributions of surface hoar samples (Samples 24 – 26) and compared them to the aggregated distributions of all other samples. We selected values of σ corresponding to the intersection of the grouped PDFs as the optimal thresholds of delineation for each dataset (i.e., each combination of instrument, illumination

condition, and incidence angle), termed σ_crit. Using these threshold values, we performed a binary pixelwise classification; pixels with σ values above σ_crit were classified as surface hoar, while values beneath were designated as "Other" microstructures. We ran the classification algorithm on all samples using the appropriate σ_crit value for each dataset. To evaluate the success of the classification algorithm, we calculated the true positive rate (TPR), true negative rate (TNR), and overall accuracy (A) for each sample using the following equations:


$$TPR = \frac{TP}{TP + FN} \qquad (3)$$

$$TNR = \frac{TN}{TN + FP} \qquad (4)$$



$$A = \frac{TP + TN}{TP + TN + FP + FN} \tag{5}$$

where TP is a true positive, TN is a true negative, etc. In this case, a TP refers to the correct identification of surface hoar when it is present, a TN corresponds to the correct identification of an "Other" microstructure, a FP is when the algorithm makes an incorrect surface hoar classification, and a FN is when surface hoar is misclassified as "Other".

## 2.5 Method repeatability assessment

Our final investigation was to test the spatial mapping capacity and repeatability of our texture-based classification on a new
snow sample, one that was not involved in the initial analysis. Using the techniques outlined in Sect. 2.1.1, we prepared a snow sample of rounded grains and grew surface hoar atop roughly *half* of the surface area, while the other half remained covered. Thus, the resulting snow surface grain shape was a 1:1 ratio of SH:RG. We proceeded to run the classification algorithm (Sect. 2.4) on the mixed sample to produce a map of surface hoar extent, using the appropriate values of $\sigma_{crit}$ for each dataset. Unfortunately, the lidar unit was no longer available at the time of this assessment, so only NIR-HSI was
evaluated. We again quantified accuracy using Equations 3 – 5.

## 3 Results

Here, we discuss results at nadir orientations, hence $\Theta = 0°$ (as defined in Fig. 3). The results of each case at $\Theta = 10°$ were very similar to their nadir counterparts, and so for clarity we address only the latter here. Results for $\Theta = 10°$ are presented in Appendix A.

### 3.1 Spatial and spectral texture analysis

We evaluated a grouped distribution of $\sigma$ for surface hoar (Samples 24 – 26) against a grouped distribution containing all other samples for each instrument/illumination condition. We performed this evaluation across a variety of spatial resolutions spanning two orders of magnitude. Further, in the case of NIR-HSI where spectral data were available, we examined results over 188 bands from 951 – 1403 nm. Using NIR-HSI under both direct and diffuse illumination, we found
that the median value of $\sigma$ for surface hoar was nearly always greater than that of "Other" microstructure group, but with considerable spatial and spectral dependence (Fig. 7a and 7b). When evaluating the difference in medians (i.e., SH minus Other), a normal distribution is evident along the spatial (vertical) axis, with the largest differences observed at the 10 mm spatial resolution for both illumination cases. Spectrally, the difference in medians remains fairly constant, peaking in the ~1250 – 1350 nm range. The maximum increase was 383% at 1246 nm for diffuse illumination and 294% at 1369 nm for
direct illumination. The differences in median values tend to be larger in the case of diffuse illumination relative to direct, and therefore we anticipate greater ease of surface hoar classification with diffuse lighting.





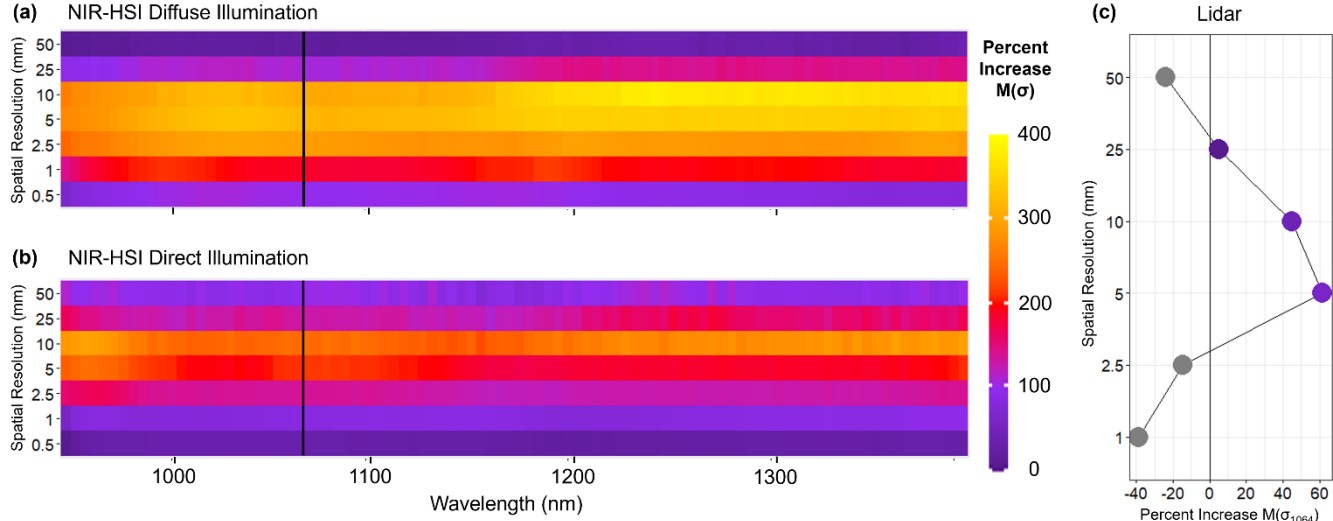

**Figure 7: Heat maps depicting the percent difference in median values of σ for surface hoar samples versus all other microstructures (SH minus other) across a variety of spatial resolutions and spectral bands (a and b). Data were acquired via NIR-HSI under diffuse (a) and direct (b) illumination conditions. In (c), a line graph illustrates the same percent difference for lidar-derived $\sigma_{1064}$ with points colored by the same scale when an increase is observed. Black vertical lines in (a) and (b) are located at the lidar wavelength of 1064 nm, for reference, while the grey vertical line in (c) is centered at zero.**

To reiterate, when using NIR-HSI, median values of σ were larger in grouped SH distributions than other microstructure distributions across nearly all cases (Fig. 7a and 7b); at worst the medians were essentially equal. This provides evidence for our hypothesis that surface hoar will produce increased NIR reflectance texture under a variety of conditions. Our lidar observations, however, were less consistent. The mere presence of heightened texture in SH was dependent on spatial resolution (Fig. 7c). At the lowest (1 mm and 2.5 mm) and highest (50 mm) spatial resolutions, we observed larger medians in the "Other" microstructure group than in SH, a result that contrasts our hypothesis. A normal distribution of delta median values corresponding to spatial resolution is evident, as in NIR-HSI, with 5 mm and 10 mm datasets producing the largest (positive) differences. As discussed, we were limited to evaluation of a single band with our lidar unit ($R_{1064}$), although our NIR-HSI spectral results (Fig. 7a and 7b) indicate that 1064 nm is a suitable wavelength.

### 3.2 Samplewise statistical analysis and significance testing

We conducted a samplewise case study using NIR-HSI derived $\sigma_{1324}$ (both direct and diffuse) at 10 mm and lidar $\sigma_{1064}$ at 5 mm spatial resolution. These selections were based on our findings in Sect. 3.1; the spatial resolutions were the optimal choice for each instrument, and 1324 nm maximized NIR-HSI texture increases under both illumination conditions. When using NIR-HSI, we found that surface hoar exhibited larger values of $\sigma_{1324}$ relative to other sample microstructures in both direct and diffuse illumination conditions. This is evident in Fig. 8d and 8e, while also outlining the problem with using NIR reflectance magnitude to delineate surface hoar. In both cases, the median reflectance magnitude ($R_{1324}$) gradually increased proceeding from lower to higher SSA (Fig. 8a and 8b), as expected. This leaves the reflectance magnitude of surface hoar



"hidden" in the middle of all other grain shapes, with moderate median $R_{1324}$ values (fuchsia horizontal reference lines). When examining values of $\sigma_{1324}$ (Fig. 8d and 8e), a different pattern is evident. In Fig. 8d and 8e, median values of $\sigma_{1324}$ are fairly constant across all samples with the exception of surface hoar, where a spike is present. The separation between the median value of $\sigma_{1324}$ for surface hoar samples compared to the best-fit line of all other samples (black lines) further illustrates this and may allow for delineation of surface hoar based on a texture parameter. Our results for lidar $R_{1064}$ indicate

a similar trend, although the distinction is less pronounced (Fig. 8c and 8f).

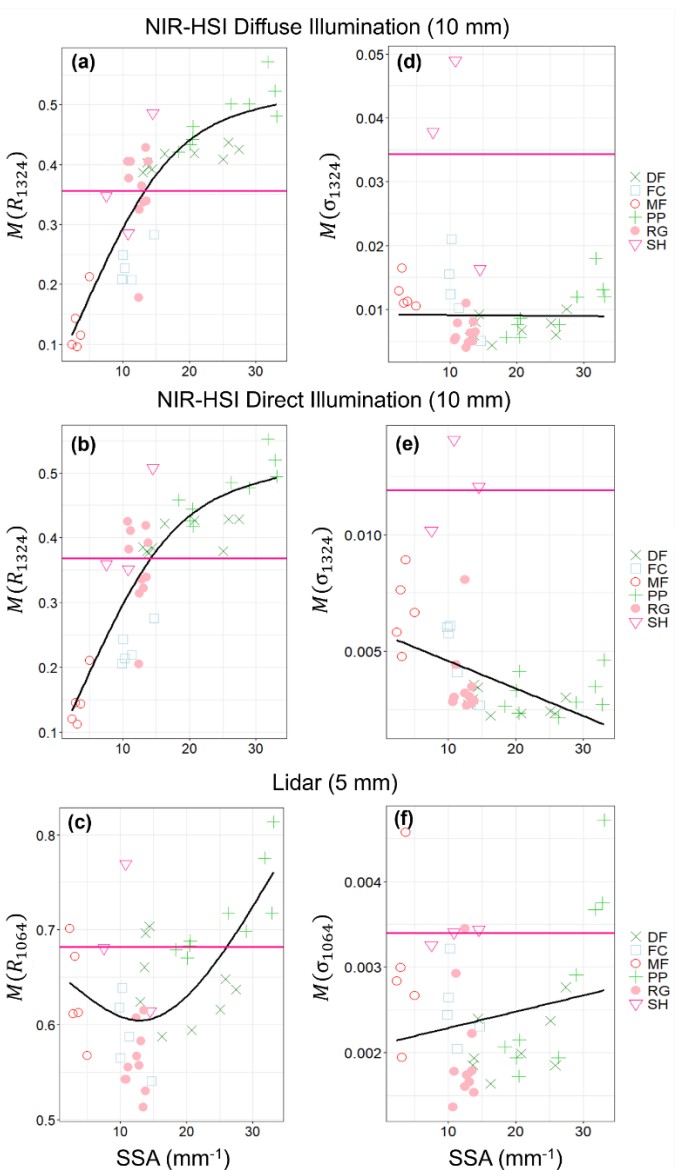

**Figure 8: Samplewise median values of reflectance (a – c) and σ (d – f) for each instrument/illumination case study. The black lines are spline or linear best fits to all samples other than surface hoar, while fuchsia horizontal reference lines depict SH median values. Colors and point shapes correspond to those suggested by the ICSSG (Fierz et al., 2009).**





To restate our central finding, when using NIR-HSI under both direct and diffuse illumination, surface hoar exhibited larger median values of $\sigma_{1324}$ relative to other sample microstructures in all cases (Table 2, Fig. 9a and 9b), thus confirming our hypothesis. Furthermore, this increase was statistically significant in all cases. The in situ Samples 25 and 26 were particularly distinct, with interquartile ranges rarely overlapping those of any other sample, making these easily discernible. Interestingly, Sample 24, which consisted of sieved SH grains (Table 1), displayed heightened texture under

direct illumination but not diffuse. As with Fig. 8, our lidar results were similar to those of NIR-HSI, but less pronounced (Fig. 9c), with particular confusion occurring with PP and MF samples. Still, the grouped SH median value of $\sigma_{1064}$ was significantly larger than sample medians in 30/38 cases. Table 2 provides complete significance findings for all scenarios, with greyed cells denoting a case where the sample median σ value is significantly less than the surface hoar median.

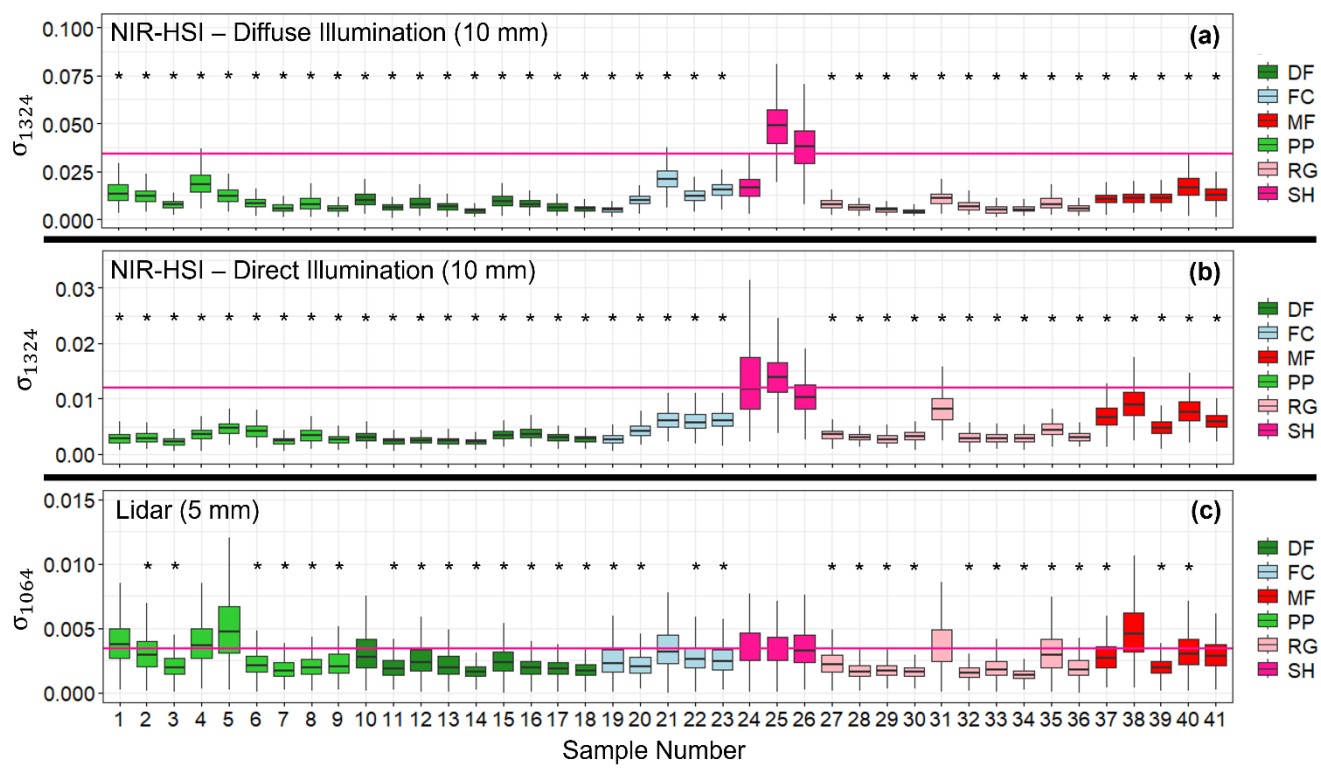


**Figure 9: Samplewise boxplot analysis for all three instrument/illumination scenarios. Boxes are colored by primary grain shape following the ICSSG. Fuchsia horizontal reference lines depict the median value of σ for surface hoar Samples 24 – 26. Samples with an asterisk had median values significantly lower than surface hoar medians based on one-sample, one-sided t-tests at α = 0.05.**






**Table 2: Sample median values of σ for each instrument and illumination condition. Median σ values of surface hoar Samples 24 – 26 are shown at the top. Sample median σ values shaded in grey denote values significantly lower than the corresponding surface hoar medians at a significance level of α = 0.05.**


| Sample # | 1° Grain Shape | NIR-HSI Diffuse | NIR-HSI Direct | Lidar |
|---|---|---|---|---|
| | | M (σ) for SH Samples 24 - 26 | | |
| | | 0.0343 | 0.0119 | 0.0034 |
| | | M ($\sigma_{1324}$) | M ($\sigma_{1324}$) | M ($\sigma_{1064}$) |
| 1 | PP | 0.0132 | 0.0027 | 0.0037 |
| 2 | PP | 0.0120 | 0.0028 | 0.0029 |
| 3 | PP | 0.0076 | 0.0022 | 0.0019 |
| 4 | PP | 0.0180 | 0.0035 | 0.0037 |
| 5 | PP | 0.0121 | 0.0046 | 0.0047 |
| 6 | PP | 0.0086 | 0.0041 | 0.0021 |
| 7 | PP | 0.0057 | 0.0024 | 0.0017 |
| 8 | PP | 0.0076 | 0.0033 | 0.0019 |
| 9 | PP | 0.0057 | 0.0026 | 0.0021 |
| 10 | DF | 0.0100 | 0.0030 | 0.0028 |
| 11 | DF | 0.0061 | 0.0023 | 0.0019 |
| 12 | DF | 0.0079 | 0.0024 | 0.0024 |
| 13 | DF | 0.0068 | 0.0023 | 0.0020 |
| 14 | DF | 0.0044 | 0.0022 | 0.0016 |
| 15 | DF | 0.0093 | 0.0034 | 0.0024 |
| 16 | DF | 0.0080 | 0.0036 | 0.0019 |
| 17 | DF | 0.0059 | 0.0029 | 0.0019 |
| 18 | DF | 0.0056 | 0.0027 | 0.0017 |
| 19 | FC | 0.0051 | 0.0027 | 0.0023 |
| 20 | FC | 0.0102 | 0.0041 | 0.0020 |
| 21 | FC | 0.0211 | 0.0061 | 0.0032 |
| 22 | FC | 0.0124 | 0.0058 | 0.0026 |
| 23 | FC | 0.0155 | 0.0060 | 0.0024 |
| 24 | SH | - | - | - |
| 25 | SH | - | - | - |
| 26 | SH | - | - | - |
| 27 | RG | 0.0081 | 0.0035 | 0.0022 |
| 28 | RG | 0.0064 | 0.0031 | 0.0017 |
| 29 | RG | 0.0049 | 0.0027 | 0.0017 |
| 30 | RG | 0.0040 | 0.0032 | 0.0016 |
| 31 | RG | 0.0111 | 0.0081 | 0.0035 |
| 32 | RG | 0.0065 | 0.0029 | 0.0015 |
| 33 | RG | 0.0050 | 0.0028 | 0.0018 |
| 34 | RG | 0.0053 | 0.0028 | 0.0014 |
| 35 | RG | 0.0079 | 0.0044 | 0.0029 |
| 36 | RG | 0.0056 | 0.0030 | 0.0018 |
| 37 | MF | 0.0105 | 0.0067 | 0.0027 |
| 38 | MF | 0.0113 | 0.0089 | 0.0046 |
| 39 | MF | 0.0110 | 0.0048 | 0.0019 |
| 40 | MF | 0.0165 | 0.0076 | 0.0030 |
| 41 | MF | 0.0129 | 0.0058 | 0.0028 |





## 3.3 Classification algorithm

We used PDFs to compare the probability density distributions of surface hoar samples (Samples 24 – 26) against all other samples, which allowed us to determine critical texture thresholds ($\sigma_{crit}$) for surface hoar delineation under each

instrument/illumination scenario. This process is illustrated in Fig. 10. Selecting the intersection point of the two distributions, following Champollion et al. (2013), allows for optimal binary classification of a given pixel. The resulting critical values for each condition are listed on the PDF plots. Using these $\sigma_{crit}$ values, we conducted pixelwise classification of all samples). Examples of this binarization for several samples of varying primary grain shape are shown in Fig. 11. All samplewise accuracy values (A) are listed in Table 3, while median values of A, TPR, and TNR for each scenario are

included on the PDF plots in Fig. 10. Classification results generally mirrored those of the statistical analysis. That is, results were excellent for NIR-HSI under both direct and diffuse illumination, but dwindled substantially when using lidar-derived $\sigma_{1064}$. The classification algorithm generally struggled most with samples of a FC or MF primary grain shape.

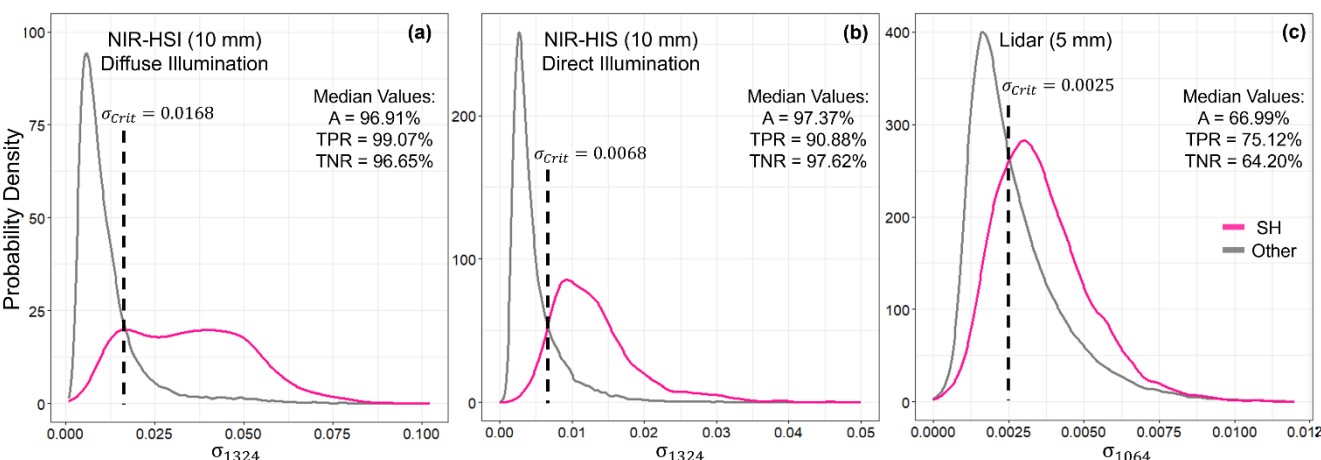

**Figure 10: Probability density functions juxtaposing the σ distributions of surface hoar, Samples 24 – 26 (fuchsia curves), with those of all other samples (grey curves). Dotted vertical reference lines represent the distribution intersection, where the critical threshold values (σ_crit) were extracted. Median accuracy metrics from the resulting samplewise binary classifications are also listed for each scenario.**







**Table 3: Samplewise and median accuracy (A) values for each instrument/illumination combination. Results for surface hoar, Samples 24 – 26, are shaded in fuchsia.**

| | NIR-HSI Diffuse | NIR-HSI Direct | Lidar |
|---|---|---|---|
| | | $\sigma_{crit}$ | |
| | 0.0168 | 0.0068 | 0.0025 |
| Sample # | | A (%) | |
| 1 | 71.67 | 99.85 | 20.26 |
| 2 | 80.81 | 98.02 | 38.31 |
| 3 | 99.48 | 100.00 | 69.69 |
| 4 | 42.19 | 95.45 | 20.89 |
| 5 | 80.67 | 91.09 | 15.60 |
| 6 | 97.27 | 93.51 | 63.71 |
| 7 | 98.62 | 100.00 | 80.09 |
| 8 | 88.01 | 99.31 | 72.02 |
| 9 | 100.00 | 100.00 | 64.68 |
| 10 | 89.50 | 97.34 | 43.93 |
| 11 | 100.00 | 100.00 | 74.31 |
| 12 | 91.18 | 100.00 | 53.45 |
| 13 | 96.91 | 100.00 | 66.99 |
| 14 | 100.00 | 100.00 | 90.67 |
| 15 | 93.35 | 97.37 | 54.83 |
| 16 | 99.15 | 94.01 | 75.43 |
| 17 | 88.97 | 98.81 | 80.64 |
| 18 | 100.00 | 100.00 | 84.12 |
| 19 | 100.00 | 99.71 | 56.27 |
| 20 | 94.16 | 95.44 | 67.89 |
| 21 | 22.71 | 66.05 | 32.27 |
| 22 | 83.87 | 68.61 | 44.99 |
| 23 | 60.34 | 65.59 | 51.89 |
| 24 | 47.76 | 86.10 | 75.12 |
| 25 | 100.00 | 97.20 | 75.97 |
| 26 | 99.07 | 90.88 | 71.58 |
| 27 | 98.04 | 95.24 | 60.70 |
| 28 | 100.00 | 100.00 | 86.56 |
| 29 | 100.00 | 99.87 | 87.04 |
| 30 | 100.00 | 97.63 | 93.47 |
| 31 | 93.83 | 31.25 | 25.54 |
| 32 | 98.66 | 97.62 | 91.21 |
| 33 | 100.00 | 100.00 | 75.55 |
| 34 | 100.00 | 98.99 | 96.40 |
| 35 | 97.30 | 89.08 | 39.34 |
| 36 | 100.00 | 99.57 | 74.68 |
| 37 | 96.39 | 52.66 | 44.08 |
| 38 | 93.42 | 22.10 | 12.32 |
| 39 | 91.74 | 85.85 | 76.60 |
| 40 | 52.24 | 37.43 | 35.75 |
| 41 | 79.27 | 72.25 | 37.70 |
| **Median** | **96.91** | **97.37** | **66.99** |



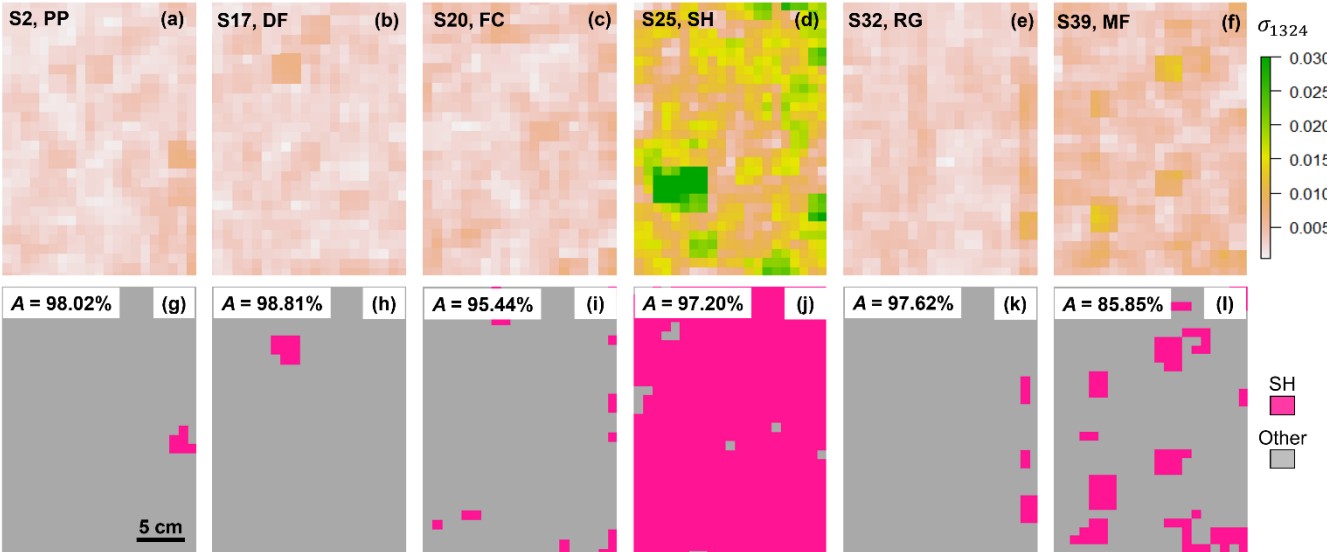

**Figure 11: Example transformations from maps of $\sigma_{1324}$ (a – f) to pixelwise classifications (g – l) via binarization with a critical threshold. The data shown here are from direct illumination at 10 mm spatial resolution.**

### 3.4 Classification assessment

Classification accuracy assessed on a new sample, comprised of a 1:1 ratio of RG and SH surface grain shapes, proved consistent, demonstrating repeatability of the texture phenomenon when using NIR-HSI (Fig. 12). Further, we observe the capacity to map surface hoar extent amid mixed surface conditions. In the $R_{1324}$ maps (Fig. 12b and 12c) the heterogeneity of surface hoar reflectance is already apparent compared to the RGs, and then quantified via a moving window analysis (Fig. 12d and 12e). Using the appropriate values of $\sigma_{crit}$ (Sect. 3.3), we binarized each texture image to create a classified data product (Fig. 12f and 12g). In the classified maps, correct rejection of the RG surface (hence TNR) was perfect (100.00%) for diffuse illumination and suitable (92.23%) under direct illumination. Accurate identification of SH (i.e., TPR) was also excellent, at 99.35% and 99.32% for diffuse and direct, respectively, while overall accuracy was 99.61% and 96.15%.



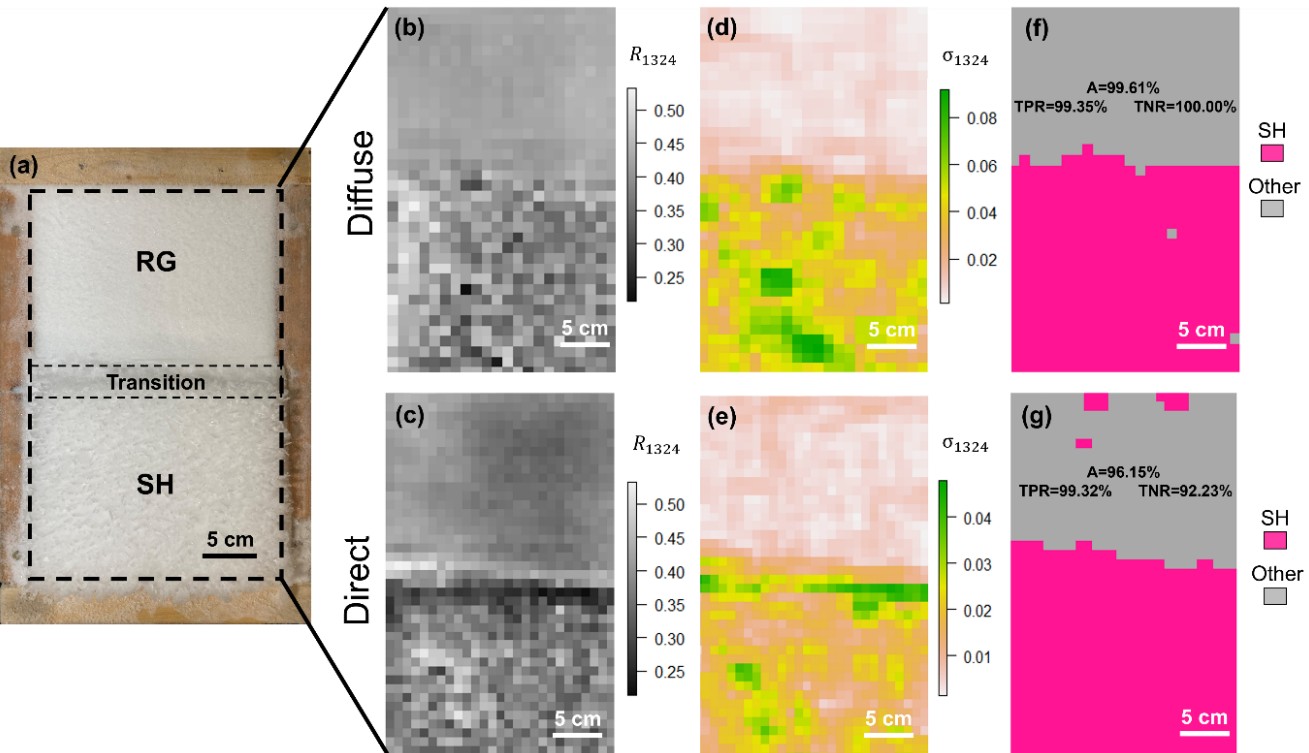

**Figure 12: A sample with a 50:50 ratio of RG:SH surface grain shapes is displayed in the visible (a). Under both diffuse (upper row) and direct (lower row) illumination conditions, the bidirectional reflectance factor at 1324 nm is extracted (b and c) from**
**NIR-HSI. Via a moving window analysis, localized standard deviation ($\sigma_{1324}$), or NIR texture, is quantified (d and e), and then binarized using critical thresholds to produce classified maps (f and g). Overall accuracy, true positive rates, and true negative rates are listed on the classified maps. The transitional zone was excluded in accuracy analyses.**

## 4 Discussion

### 4.1 NIR-HSI

We found that, when measured with NIR-HSI, surface hoar exhibited larger median values of $\sigma$, a NIR texture metric, than other snow microstructures across a variety of spatial resolutions and spectral bands. Furthermore, in a samplewise case study evaluating $\sigma_{1342}$ at 10 mm spatial resolution, we determined that median surface hoar values were significantly larger than other snow surface structures (primary grain habits of PP, DF, FC, RG and MF) in all cases under both direct and diffuse illumination (one-sided, one-sample t-test, $\alpha = 0.05$). Our findings are consistent with Champollion et al. (2013), who
found that under artificial lighting conditions the NIR texture (in this case a contrast index) of the Antarctic snow surface was higher when surface hoar was present as compared to when it was absent. These researchers used a passive NIR camera and downward-looking lights, all mounted ~ 2 m above a flat snow surface, to create a field setup akin to the direct illumination ($\Theta = 0°$) case presented here. Although they do not explicitly mention the spatial resolution of their images, they imply that individual surface hoar crystals spanned 5 – 10 pixels, and thus their resolution was likely on the order of



millimeters. While Champollion et al. (2013) simply juxtaposed the presence versus absence of surface hoar, we build on these foundational findings by quantifying the texture phenomenon against a variety of differing, thoroughly characterized snow microstructures in a controlled laboratory environment. Further, our work features the explicit inclusion of diffuse illumination, varied illumination/viewing incidence angle, and a range of spatial resolutions and spectral bands. Last, whereas Champollion et al. (2013) used texture metrics across an entire image to confirm the presence of hoar crystals on a

given day (with accuracy of 94%), we extend this classification by demonstrating a per-pixel mapping methodology, allowing us to map the spatial extent of surface hoar within an image (Fig. 12). Median sample classification accuracy was 96.91% and 97.37% for diffuse and direct illumination, respectively. As discussed in Sect. 3.1 and illustrated in Fig. 8, surface hoar exhibits moderate NIR reflectance when compared to other grain shapes, and thus leveraging reflectance magnitude alone is insufficient for surface hoar delineation. These results highlight the importance of our texture-based

approach.

        Spatial resolution was perhaps the factor with the most influence on our results, with the maximal difference between surface hoar and other median values occurring at 10 mm resolution for both illumination cases (Fig. 7). However, consistent differences in median values can be observed at nearly all resolutions beneath 50 mm, indicating that classification at coarser resolutions may still be suitable under the right conditions. This trend makes sense physically; we

attribute the rise in texture metrics associated with surface hoar to be related to the lateral spacing between large hoar crystals (Fig. 1). Given that this spacing is often on the order of millimeters, it is possible that the raw resolution is too fine to optimally observe this variability, while 50 mm resolution is too coarse. Further, the fact that results were fairly consistent spectrally from 951 – 1403 nm implies that hyperspectral capacity is likely unnecessary; a broadband passive NIR sensor could likely observe the same texture increases. Samples that proved the most challenging to discern from SH were FC

(Samples 19 – 23), and the large MF grains from Batch L (Samples 37 – 41). The latter was likely due to enhanced surface roughness, as these grains were several mm in length (Fig. 2), a rather extreme case, and indicates that caution should be used when conditions favor wet snow metamorphism. Sporadic misclassification of FC (e.g., Fig. 11i) is perhaps understandable, given that these grains form from kinetic snow metamorphism, which is somewhat similar to surface hoar growth. In practice, it may be useful to try to identify these surfaces as well, as near-surface FC also tend to act as weak

layers once buried. The relatively lower median $\sigma_{1342}$ value of the sieved SH Sample 24 under diffuse illumination (Fig. 9a) is an interesting result. This anomaly perhaps implies that the texture signature dissipates when the predominately vertical orientation of SH grains is interrupted, and thus surface hoar that has been blown over is likely more difficult to detect. The low accuracy value of this sample under diffuse illumination (47.76%, Table 3) is evidence of this. The uncertainty involved in SH classification for any case can be quantified by examining the area under the intersecting PDF curves (Fig. 10).

Furthermore, uncertainty with regards to a specific microstructure (such as FC) could be observed by comparing PDFs between individual samples, rather than grouping SH samples versus all other samples. However, our goal was to determine robust thresholds for delineation of SH from any other microstructure.



## 4.2 Lidar

For as impressive as the performance of NIR-HSI was, the results of our lidar analysis were more perplexing. In Fig. 7c, we
can observe that the median value of $\sigma_{1064}$ for SH Samples 24 – 26 was only greater than that of other microstructures at spatial resolutions of 5, 10, and 25 mm. At the largest and smallest resolutions, much like with NIR-HSI, performance diminished. However, unlike NIR-HSI, in these cases the median SH value was actually *lower* than the other microstructure median, a result that runs counter to our hypothesis. Even at 5 mm resolution, our lidar classification results were considerably less robust than those of NIR-HSI, with median sample accuracy of 66.99%. Unexpected results in texture
analyses are not unheard of. While Champollion et al. (2013) noted increases in NIR texture when surface hoar was present under direct, "artificial" illumination at nadir incidence (as we did with NIR-HSI), they also documented a reversal of this trend under natural (solar) illumination. However, this was at very large (80° - 85°) solar zenith angles, and thus represents an extreme case of grazing incidence, and the cause of this observation was not thoroughly explained. For the laboratory setup used here, we expected lidar to reproduce similar results to those of our NIR-HSI analysis and to the findings of
Champollion et al. (2013) using artificial light. Like NIR-HSI, the lidar classification algorithm struggled with the large MFs of Batch L, as well as PP (Fig. 9c).

In general, the use of lidar reflectance to ascertain snow surface properties is far less validated relative to passive imagery, like NIR-HSI. While passive NIR imagery has been used to estimate SSA or $r_e$ for decades, leveraging the well-established sensitivity of NIR reflectance to snow microstructure, very few studies (e.g., Yang et al., 2017) have attempted to
do so with lidar. Therefore, the dependence of NIR lidar reflectance on snow microstructure is substantially less understood. This is important because lidar scanning represents a unique bidirectional reflectance scenario. For instance, the beam emitted from lidar units is collimated, and thus it experiences less loss due to scattering and absorption compared to direct solar irradiance, and results in higher irradiance at the wavelength of interest. Collimation alters scattering and reflectance in optically rough materials like snow relative to an un-collimated illumination source (Murphy, 2006). Additionally, lidar
beams are predominately linearly polarized, which contrasts with the often randomly polarized nature of solar illumination (Sassen, 2005). The scattering process is partly polarizing, and therefore multiple scattering can have significant polarization dependence (Li et al., 2008; Bhandari et al., 2011). Last, lidar presents a monostatic geometric condition on bidirectional reflectance, such that every measurement strictly observes direct backscatter. This is beneficial in that it limits the number of characterizations required compared to a full bidirectional reflectance investigation. However, observations of snow
reflectance at this geometry are lacking, and radiative transfer models are not well-validated for the case of direct backscatter. Thus, lidar is an optically unique case of bidirectional reflectance that requires careful examination, so it is not entirely surprising that a texture analysis of lidar reflectance produced peculiar results.

One possible explanation for the relatively poor performance of our algorithm when using lidar reflectance can be found in the work of Walter et al. (2023), who leveraged a laboratory setup much like the apparatus presented here. These
researchers focused on a temporal analysis, using a 905 nm lidar to observe changes in the mean reflectance magnitude and



standard deviation (across an entire image) at prescribed time intervals as they grew surface hoar atop a layer of compressed PP. Consistent with our spatial analysis, they noted that reflectance magnitude was insufficient to characterize surface hoar growth, as the mean reflectance changed only 4%, while standard deviation increased as much as 600%. Although this juxtaposition features only a single microstructure other than SH (compressed PP), these lidar-based results seem more

encouraging than our own, and are more consistent with our NIR-HSI findings. A key distinction is, perhaps, the lidar spot size. Their spot size of 3 mm is much smaller than that of our lidar/setup (15 mm), and likely closer to the size and spacing of individual surface hoar crystals in many cases. Indeed, the authors predict that surface hoar crystals smaller than the lidar beam diameter will not be detectable, and thus it is possible that our beam diameter is simply too coarse for this application.

## 4.3 Future work and speculations on scaling to field applications

Though scaling to field applications presents considerable challenges and uncertainty, it is likely that our NIR-HSI findings can be extended to operational avalanche forecasting in the near future. A simple setup like that of Champollion et al. (2013), where a downward-looking NIR camera acquired daily and nightly images, could be installed at remote weather stations to monitor the formation and persistence of surface hoar layers prior to burial. Such remote measurements are not currently available but are critically important for avalanche forecasters. Further, UAV snow mapping using NIR-HSI has

recently been demonstrated as a useful tool to measure snow grain size and albedo at the slope scale (Skiles et al. 2023) and future studies should consider texture analyses. To accomplish this, a greater variety of incidence angles must be evaluated. Although it is encouraging that our results remained promising at $\Theta = 10°$ (Appendix A), more oblique angles will inevitably be encountered in the field. Even on overcast days, where diffuse solar illumination is prevalent, variance in terrain slope beneath a downward looking imager would still alter the viewing incidence angle and thus the magnitude, and likely texture,

of NIR snow reflectance. Using direct solar illumination would add another factor, the illumination incidence angle, although this could be kept consistent by using artificial lighting. Ideal thresholds of $\sigma_{crit}$ will likely continue to vary between incidence angles, as well as between instruments/applications. Further, the flight plan and/or optical logistics would need to be controlled to ensure adequate spatial resolution. Another factor that may be worth investigating is the window size during focal analysis, although a preliminary study determined that the neighborhood size was of little consequence for our data.

Although we realized limited success with lidar, the use of lidar for surface hoar mapping via texture requires further evaluation, particularly with regards to beam diameter. A better physical understanding regarding the resolution-dependence of our lidar results is needed. At a minimum, lidar could be useful in conjunction with NIR-HSI or other passive NIR detection. Using the spatial capacity of lidar, values of slope angle (and thus incidence angle if using a downward-oriented imager), could be determined on a per-pixel basis, allowing for the proper threshold value to be used when

analyzing NIR-HSI texture data. Though challenging, slope, or even basin-scale mapping of surface hoar extent is likely possible with current technology, though a field campaign would be required to tune our method to a wider variety of illumination conditions and incidence angles. Maps of surface hoar extent over avalanche terrain prior to burial would be vital in making slope-specific avalanche forecasts. Such maps could also identify likely avalanche trigger points, improving





avalanche mitigation with explosives, for example. While our work focused on surface hoar classification, texture analysis
will likely provide a useful tool for evaluating other physical phenomena as well, such as surface roughness.

**5 Conclusion**

Our research demonstrates a novel method for mapping the spatial extent of surface hoar using NIR texture in a cold
laboratory. In essence, we found that:

I.    Hyperspectral imaging can robustly measure the texture of snow and ice by computing pixelwise variability in
550        reflectance at any NIR wavelength.

II.   When evaluated with NIR-HSI, surface hoar has significantly heightened NIR texture relative to other snow
         microstructures across a range of spectral bands and spatial resolutions, likely as a result of variable ice absorption
         and specular contributions.

III.  Near-infrared texture thresholds can be used to binarize NIR-HSI texture measurements, resulting in accurate maps
555        of surface hoar spatial extent on a per-pixel basis.

IV.   Similar results were achieved with 1064 nm lidar, although the phenomenon was resolution-dependent and
         performance was substantially less robust. The use of lidar for this purpose requires further investigation, and is
         likely dependent on beam diameter.

As NIR-HSI and lidar become more economical, these may provide capable methods for measuring NIR texture and
subsequently mapping surface hoar. Extending the work presented here to field operations will have immediate implications
for broad-scale snow surface mapping and avalanche forecasting.







# 6 Appendix A

Results for the Θ = 10° incidence viewing angle case are presented below.

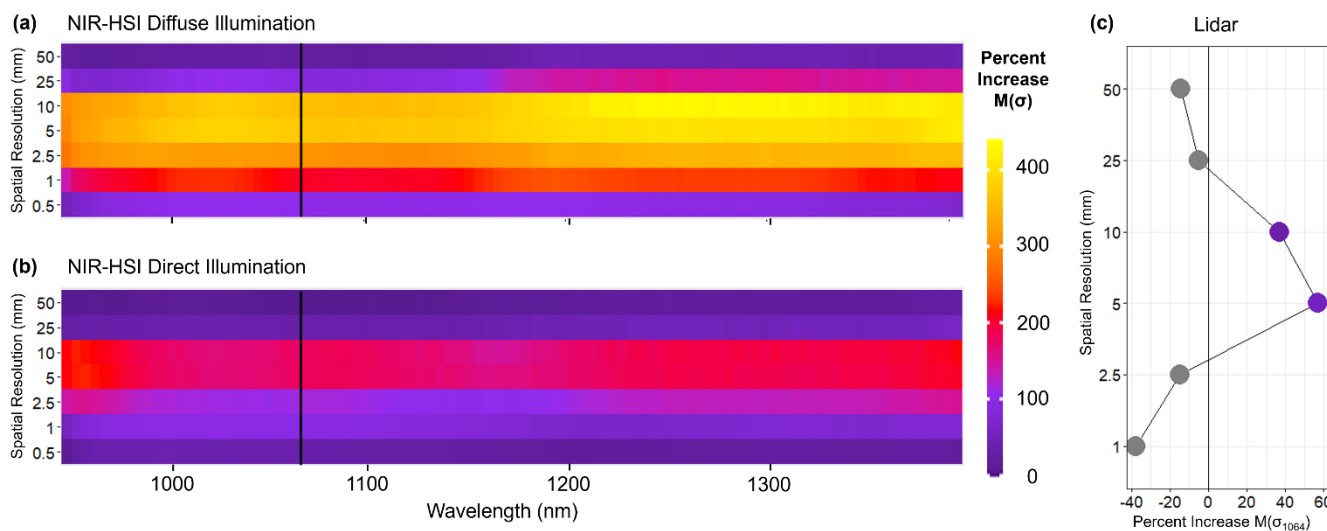

**Figure A1: Heat maps depicting the percent difference in median values of σ for surface hoar samples versus all other microstructures (SH minus other) across a variety of spatial resolutions and spectral bands (a and b). Data were acquired via NIR-HSI under diffuse (a) and direct (b) illumination conditions. In (c), a line graph illustrates the same percent difference for lidar-**
**derived σ_{1064} with points colored by the same scale when an increase is observed. Black vertical lines in (a) and (b) are located at the lidar wavelength of 1064 nm, for reference, while the grey vertical line in (c) is centered at zero. Relative to the nadir case, performance generally improved for NIR-HSI under diffuse illumination, but decreased under direct illumination for both NIR-HSI and lidar.**








**Figure A2: Samplewise median values of reflectance (a – c) and σ (d – f) for each instrument/illumination case study. The black lines are spline or linear best fits to all samples other than surface hoar, while fuchsia horizontal reference lines depict SH median values. Colors and point shapes correspond to those suggested by the ICSSG (Fierz et al., 2009).**



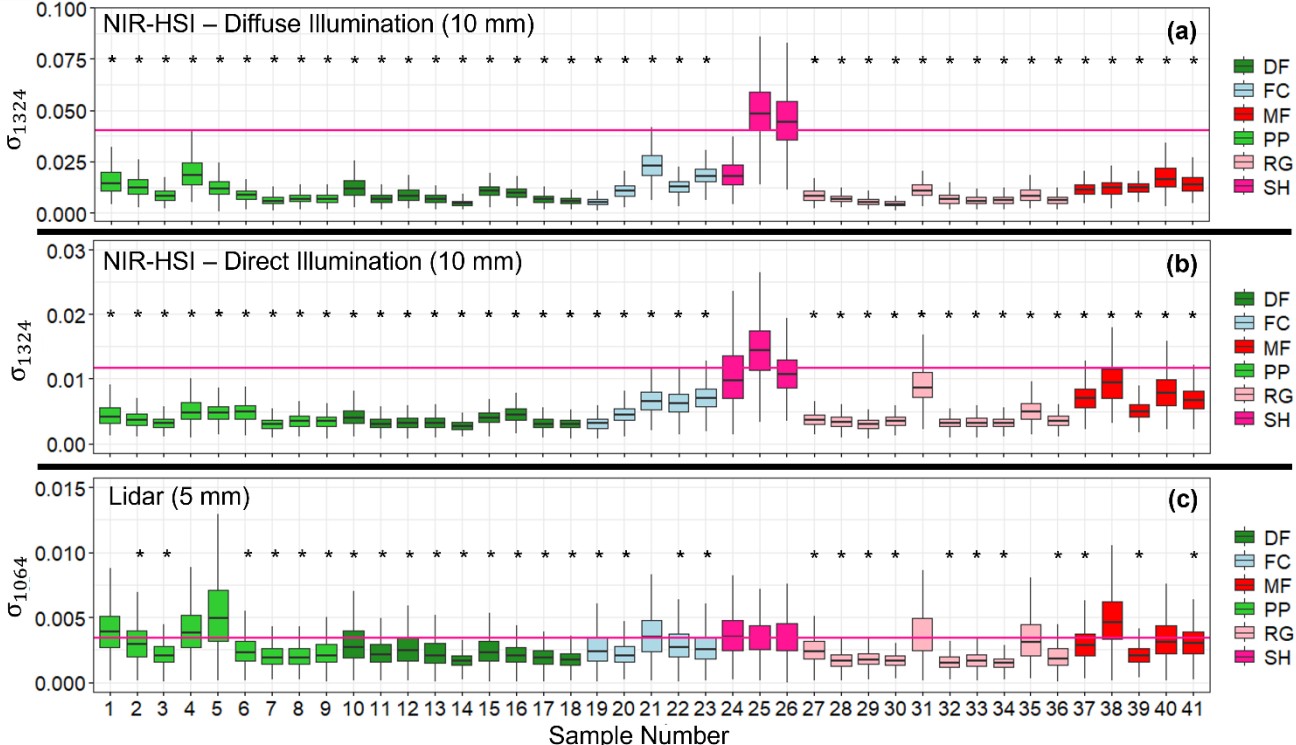

**Figure A3: Samplewise boxplot analysis for all three instrument/illumination scenarios. Boxes are colored by primary grain shape following the ICSSG. Fuchsia horizontal reference lines depict the median value of σ for surface hoar Samples 24 – 26. Samples with an asterisk had median values significantly lower than surface hoar medians based on one-sample, one-sided t-tests at α = 0.05.**



**Table A1: Sample median values of σ for each instrument and illumination condition. Median σ values of surface hoar Samples 24 – 26 are shown at the top. Sample median σ values shaded in grey denote values significantly lower than the corresponding surface hoar medians at a significance level of α = 0.05.**

| Sample # | 1° Grain Shape | NIR-HSI Diffuse | NIR-HSI Direct | Lidar |
|---|---|---|---|---|
| | | M (σ) for SH Samples 24 - 26 | | |
| | | 0.0400 | 0.0117 | 0.0034 |
| | | M ($\sigma_{1030}$) | M ($\sigma_{1030}$) | M ($\sigma_{1064}$) |
| 1 | PP | 0.0145 | 0.0042 | 0.0039 |
| 2 | PP | 0.0123 | 0.0036 | 0.0029 |
| 3 | PP | 0.0084 | 0.0032 | 0.0021 |
| 4 | PP | 0.0186 | 0.0047 | 0.0038 |
| 5 | PP | 0.0119 | 0.0048 | 0.0049 |
| 6 | PP | 0.0089 | 0.0049 | 0.0023 |
| 7 | PP | 0.0059 | 0.0030 | 0.0019 |
| 8 | PP | 0.0069 | 0.0034 | 0.0019 |
| 9 | PP | 0.0066 | 0.0034 | 0.0021 |
| 10 | DF | 0.0117 | 0.0040 | 0.0027 |
| 11 | DF | 0.0067 | 0.0030 | 0.0021 |
| 12 | DF | 0.0084 | 0.0032 | 0.0024 |
| 13 | DF | 0.0066 | 0.0032 | 0.0021 |
| 14 | DF | 0.0047 | 0.0027 | 0.0016 |
| 15 | DF | 0.0105 | 0.0040 | 0.0023 |
| 16 | DF | 0.0097 | 0.0044 | 0.0020 |
| 17 | DF | 0.0065 | 0.0030 | 0.0019 |
| 18 | DF | 0.0058 | 0.0031 | 0.0017 |
| 19 | FC | 0.0053 | 0.0031 | 0.0024 |
| 20 | FC | 0.0106 | 0.0045 | 0.0021 |
| 21 | FC | 0.0229 | 0.0065 | 0.0035 |
| 22 | FC | 0.0125 | 0.0063 | 0.0027 |
| 23 | FC | 0.0180 | 0.0070 | 0.0026 |
| 24 | SH | - | - | - |
| 25 | SH | - | - | - |
| 26 | SH | - | - | - |
| 27 | RG | 0.0082 | 0.0036 | 0.0024 |
| 28 | RG | 0.0066 | 0.0033 | 0.0016 |
| 29 | RG | 0.0053 | 0.0030 | 0.0017 |
| 30 | RG | 0.0044 | 0.0035 | 0.0016 |
| 31 | RG | 0.0109 | 0.0087 | 0.0034 |
| 32 | RG | 0.0065 | 0.0032 | 0.0015 |
| 33 | RG | 0.0057 | 0.0032 | 0.0016 |
| 34 | RG | 0.0060 | 0.0032 | 0.0015 |
| 35 | RG | 0.0082 | 0.0049 | 0.0031 |
| 36 | RG | 0.0061 | 0.0035 | 0.0018 |
| 37 | MF | 0.0113 | 0.0070 | 0.0028 |
| 38 | MF | 0.0121 | 0.0094 | 0.0046 |
| 39 | MF | 0.0120 | 0.0050 | 0.0021 |
| 40 | MF | 0.0161 | 0.0078 | 0.0031 |
| 41 | MF | 0.0136 | 0.0066 | 0.0030 |





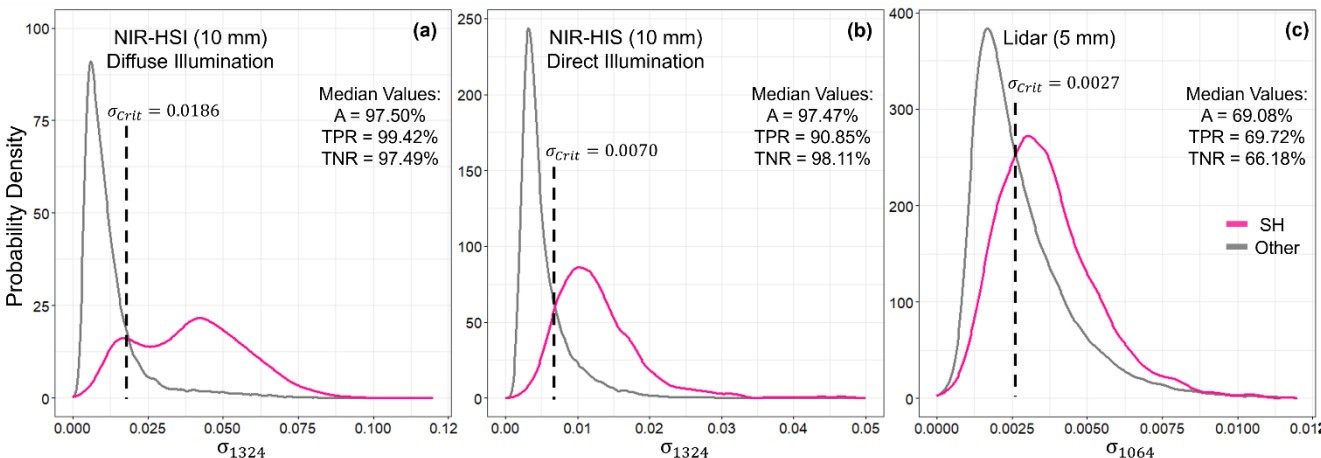

**Figure A4: Probability density functions juxtaposing the σ distributions of surface hoar, Samples 24 – 26 (fuchsia curves), with those of all other samples (grey curves). Dotted vertical reference lines represent the distribution intersection, where the critical threshold values (σ_crit) were extracted. Median accuracy metrics from the resulting samplewise binary classifications are also listed for each scenario.**





**Table A2: Samplewise and median accuracy (A) values for each instrument/illumination combination. Results for surface hoar, Samples 24 – 26, are shaded in fuchsia.**

| | NIR-HSI Diffuse | NIR-HSI Direct | Lidar |
|---|---|---|---|
| | | $\sigma_{crit}$ | |
| | 0.0186 | 0.0070 | 0.0027 |
| Sample # | | A (%) | |
| 1 | 72.33 | 89.15 | 24.87 |
| 2 | 83.13 | 95.96 | 45.25 |
| 3 | 99.68 | 99.23 | 73.05 |
| 4 | 50.38 | 83.29 | 24.38 |
| 5 | 85.71 | 90.30 | 18.42 |
| 6 | 99.23 | 90.61 | 63.28 |
| 7 | 99.27 | 100.00 | 77.67 |
| 8 | 97.84 | 97.86 | 78.17 |
| 9 | 99.68 | 100.00 | 70.60 |
| 10 | 85.13 | 93.71 | 51.30 |
| 11 | 100.00 | 99.47 | 69.40 |
| 12 | 92.55 | 99.79 | 58.86 |
| 13 | 97.50 | 99.52 | 69.08 |
| 14 | 100.00 | 99.86 | 92.61 |
| 15 | 95.87 | 98.92 | 62.54 |
| 16 | 99.56 | 93.72 | 75.56 |
| 17 | 99.71 | 98.88 | 83.55 |
| 18 | 99.72 | 100.00 | 87.14 |
| 19 | 100.00 | 99.03 | 59.43 |
| 20 | 97.30 | 92.69 | 71.82 |
| 21 | 26.82 | 59.36 | 31.77 |
| 22 | 91.19 | 65.53 | 49.82 |
| 23 | 54.19 | 49.75 | 54.41 |
| 24 | 45.48 | 75.78 | 69.72 |
| 25 | 99.49 | 97.47 | 71.65 |
| 26 | 99.42 | 90.85 | 68.56 |
| 27 | 97.47 | 98.79 | 61.54 |
| 28 | 99.85 | 99.88 | 89.85 |
| 29 | 100.00 | 99.75 | 90.04 |
| 30 | 100.00 | 99.55 | 95.68 |
| 31 | 96.31 | 23.70 | 31.09 |
| 32 | 99.32 | 100.00 | 90.86 |
| 33 | 100.00 | 98.37 | 86.31 |
| 34 | 98.78 | 99.88 | 98.35 |
| 35 | 97.39 | 85.68 | 41.07 |
| 36 | 100.00 | 98.51 | 77.23 |
| 37 | 96.67 | 49.46 | 46.40 |
| 38 | 91.27 | 25.22 | 14.25 |
| 39 | 97.30 | 86.69 | 77.62 |
| 40 | 61.71 | 41.65 | 39.89 |
| 41 | 81.77 | 56.24 | 39.62 |
| **Median** | **97.50** | **97.47** | **69.08** |




## 7 Data availability

Data will be made available upon request from the lead author.

## 8 Author contribution

JD conceptualized the study, collected laboratory data, and analyzed results. CD provided guidance and advised during conceptualization and especially analysis. ES was instrumental with laboratory data collection. KB provided conceptual guidance, particularly during earlier iterations of the project. KH acquired funding for this research, was responsible for project administration, provided conceptual guidance, and supervised JD throughout the study. JD wrote the original draft manuscript, and all co-authors contributed during review and editing.

## 9 Competing interests

The authors declare that they have no conflict of interest.

## 10 Acknowledgements

This work was funded by the Transportation Avalanche Research Pooled Fund Program (TARP), administered through the Colorado Department of Transportation (CDOT), and by NASA Grant 80NSSC22K0694 from the Terrestrial Hydrology Program. We acknowledge the services and equipment (Riegl VZ-6000) provided by the GAGE Facility, operated by 655 UNAVCO, Inc., with support from the National Science Foundation and the National Aeronautics and Space Administration under NSF Cooperative Agreement EAR-1724794. We acknowledge the use of the Subzero Research Laboratory in the Department of Civil Engineering at Montana State University and thank Ladean McKittrick for laboratory assistance. We would like to thank Resonon, Inc. for providing us with a hyperspectral imager and technical assistance. Last, we thank Joseph Shaw, Nathaniel Field and Riley Logan for technical guidance regarding optical data acquisition.

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
