# Peer review of "Mapping surface hoar from near-infrared texture in a laboratory"

_EGUsphere, 2023_

## Author Response (AR1)

To Dr. Alexander Kokhanovsky:

This paper is aimed at the development of the technique for mapping snow surface hoar using near IR reflectance texture. The subject is very important for many applications. I would suggest that the paper is published as it stands.

Thank you for taking the time to review our paper. We are pleased to hear that you find the topic of mapping snow surface hoar to be important and that you believe our paper is suitable for publication in its current form. Your endorsement means a great deal to us, and we appreciate your support.

To anonymous reviewer 2:

We thank the reviewer for their kind words, support, and thorough feedback. See below for our responses to specific comments.

**L28: This accuracy is only for NIR-HIS, no? Maybe worth mentioning.**

That is correct. We shall revise this line to read, "…with accuracy of 99.61% and 96.15% for NIR-HSI under direct and diffuse illumination, respectively."

**Table 1: PP particles have a quite low SSA. How old were the samples? The low SSA is likely a result of the low micro-CT resolution you used for the measurements (15um) as you miss the fine dendrites <15um. Your SSA may be significantly higher.**

The samples were freshly made or collected. This is a good point, and we agree that the relatively coarse micro-CT resolution likely resulted in an underestimation of SSA for PP samples (particularly S1 – 5). This was the minimum resolution achievable with the relatively large cylindrical micro-CT sample holder used in this study (Fig. 3d). The larger micro-CT sample holder was chosen to provide sufficient surface area for larger-grained samples, namely surface hoar, to be encapsulated and transported to micro-CT for measurement. We will acknowledge this uncertainty in Section 2.1.2, where we discuss micro-CT and will add the following statement:

"The voxel size of 14.5 µm was the finest spatial resolution achievable with the relatively large cylindrical micro-CT sample holder used in this study. The larger micro-CT sample holder was chosen to provide sufficient surface area for larger-grained samples, namely surface hoar, to be encapsulated and transported to micro-CT for measurement. We intuitively note that this will result in neglection or mischaracterization of fine dendrites smaller than this size. Consequentially, it is possible that the SSA of samples with a PP primary grain habit (particularly Samples 1 – 5) was underestimated by micro-CT, although this is of little consequence for the optical texture analysis presented here."

**L193: Spectralon panel: This is not clear, but I guess you placed the spectralon reference array around the snow sample? How did you correct for inhomogeneous illumination of the snow sample? Did you make a reference measurement with a homogeneous white plate? Maybe you can add a few sentences to discuss this?**

Yes, we can further clarify this. Our spectralon panel is 30.5 x 30.5 cm, thus larger in both dimensions than our snow optical ROI. We built a sample holder with the same external dimensions as our snow sample holders, but specifically made to hold the spectralon panel, both centered on the ROI and at the same distance from the illumination source as the snow surfaces. For each snow sample scan, we also conducted a reference scan with the spectralon panel. This allowed for pixel-by-pixel calibration of the entire optical ROI, thus accounting for any heterogeneous illumination. We made these reference measurements for each sample and each illumination condition. We will add this description to Section 2.2.1.1, where we first introduce calibration via the spectralon panel, and reference it again in the diffuse section near L193.

**L205: finer = lower?**

What we meant was that, by coarsening our laboratory samples, we approach the finer end of spatial resolutions achievable by UAV applications. We can see how this wording is confusing. Since it adds very little, we've elected to simply remove the word, "finer", such that the line reads, "…as an attempt to mimic the spatial resolutions achievable by UAV-mounted systems." Essentially the same statement, hopefully with less confusion.

**L272+: Chosen wavelength 1324nm -> is also within the wavelength range where the effect of optical equivalent diameter on reflectance is maximum. Likely also maximizes texture? Maybe provide some information on the size of the surface hoar crystals and on what substrate they are grown and how the substrate may affect the texture?**

Yes, this wavelength is sensitive to optical grain size or the path length of ice and is therefore likely the reason why we see that the texture is maximized in this region. We will highlight this by adding the following statement at line ~L274:

"…indicated that this was an optimal wavelength under both direct and diffuse illumination. This is a sensible finding, considering that 1324 nm sits amid a prominent ice absorption feature where reflectance is particularly sensitive to the path length of ice (i.e., optical grain size) at this wavelength."

The SH crystals were ~0.5 – 2.0 cm in length. Example images of individual crystals on a 2 mm grid are shown in Fig. 2, but we will explicitly state this in Section 2.1.1. The SH grains were grown atop RGs, which is mentioned in Section 2.1.1. (Line 134).

We do not expect the substrate grain habit to have a substantial effect on texture. Further, we measured the texture of the substrate, RGs, via samples 27 – 36, which had a RG primary grain habit, and found it to be minimal (e.g., Fig. 9). However, varying the underlying grains would certainly be interesting, and there is probably some influence, especially with smaller SH crystals. In retrospect, that would have been a useful sample addition. We shall suggest this in Section 4.3 where we discuss future work.

**Fig. 7c: Decreasing spatial Lidar resolutions below 15 mm becomes increasingly irrelevant with a beam size of 15mm. But it is clear what you mean. Maybe just quickly discuss this fact somewhere.**

Yes, this is a good point. We can add a line reminding the reader of this fact when we first discuss coarsening the lidar maps in Section 2.2.2.2, L226. For instance, "…to examine the influence of spatial resolution on NIR texture. However, we remind the reader that the original lidar beam diameter is 15 mm, and thus spatial resolutions lower than this are not as relevant as with NIR-HSI."

**Fig. 10: If you have PDF on the y-axis for all three plots, shouldn't it be 0-100% or 0-1 for all? If you have "number of samples/subarrays" then the total number of SH is maybe too high relative to "others"? Something seems wrong, if not, maybe some better description is needed here.**

A probability density function describes the *relative* likelihood of sampling a certain value. Typically, the y-values are designated such that the area under each curve is equal to 1. So, the y-values are dependent on the magnitude of the x-values, which in this case are quite small and vary between the three plots, hence the differing y-axis values. We will add a line in the Fig. 10 caption better clarifying the meaning of the y-axis values, and/or normalize the y-values so that the reader can more easily interpret this as a probability.

**L461: Too fine resolution means that the 3x3 window does not detect any variability as the entire subarray may be located e.g. on a single SH crystal?**

Precisely, that is our assumption anyway. We note later (L533) that the neighborhood size is a topic of interest for future research. Although we performed a preliminary analysis on this factor and didn't notice much difference, intuitively there must be some relevant interplay between spatial resolution and window size.

**L463: Maybe worth mentioning that standard CCD cameras detect light up to 950nm when removing the NIR filter. These cameras are much cheaper compared to real NIR InGaAs cameras.**

Good point, we'll add a reference to this on L464. The lidar work of Walter et al. (2023), that we discuss at L509+, also provides optimism for lower NIR wavelengths, as they use a 905 nm lidar.

**Check for different writing of the word Lidar or lidar.**

Completed. The lowercase form is used everywhere except for where the word begins a sentence or is a section/table heading.

**Appendix: Maybe quickly discuss the reason why 10° results are similar to the 0° results … BRDF. Fig. A1b shows stronger differences to Fig. 7b for direct illumination compared to A1a/7a for diffuse illumination. Maybe because the vertically aligned SH crystals result in reduced sigma under direct illumination at an angle of 10°? A short discussion of this and maybe other small/minor differences (few sentences) at the beginning of the Appendix would be nice.**

We will extend the introductory sentence at the beginning of the appendix to a full introductory paragraph describing the similarities/differences:

"Results for the $\Theta$ = 10° incidence viewing angle case are presented below. As mentioned in the text, the results are very similar to the case of $\Theta$ = 0°, particularly regarding the samplewise comparison and classification mapping. This is perhaps unsurprising, as the bidirectional reflectance distribution function (BRDF) of snow changes very little for the case of direct backscatter between 0 – 10°. Thus, we certainly would not expect large changes in reflectance magnitude, and this is likely true for spatial variability as well. We observed the most pronounced differences in the spatial/spectral analysis. By juxtaposing Fig. 7 and Fig. A1, we note slightly larger increases in M($\sigma$) under diffuse illumination for $\Theta$ = 10° relative to $\Theta$ = 0°, while the opposite is true under direct illumination. The latter difference is likely related to how non-nadir direct illumination interacts with the predominately vertically oriented SH crystals, although this topic requires further investigation. As mentioned in Sect. 4.3, it is imperative that future studies evaluate texture at more oblique incidence angles, particularly when considering scaling to field applications."

**Maybe you want to add these references:**

**Chandel, C., Srivastava, P. K., Kumar, V., Datt, P., Sheoran, R. and Satayawali, P. K. (2023) Laboratory set-up for surface hoar layer growth over rounded grain snow. Cold Regions Science and Technology 205**

**Ozeki, T., Tsuda, M., Yashiro, Y., Fujita, K. and Adachi, S. (2020) Development of artificial surface hoar production system using a circuit wind tunnel and**

**formation of various crystal types. Cold Regions Science and Technology 169, https://doi.org/10.1016/j.coldregions.2019.102889.**

These are both quite similar to our apparatus. We will include these references in Sect. 2.1.1, where we discuss sample preparation.

In response to the preceding review file validation:

**1) Checking your paper, I have noticed that your tables contain coloured cells. Please note that this will not be possible in the final revised version of the paper due to HTML conversion of the paper. When revising the final version, you can use footnotes or italic/bold font. For now, the process will continue, but please note that the final version cannot be published by using coloured tables. 2) Please ensure that the colour schemes used in your maps and charts allow readers with colour vision deficiencies to correctly interpret your findings (see e.g. F09). Please check your figures using the Coblis Color Blindness Simulator (https://www.color-blindness.com/coblis-color-blindness-simulator/) and revise the colour schemes accordingly.**

1) We replaced the table shading with either bold or italicized fonts, depending on the table.

2) We were unable to access the Coblis simulator (some sort of authorization/security issue), so we instead utilized the simulator available on pilestone.com. It looks like Fig. 9 should be reasonably resilient to most color blindness with the exception of monochromacy/achromatopsic. While we do recognize the importance of choosing appropriate color schemes and we applaud this initiative, we would like to point out that the color scheme selected in Fig. 9 was not arbitrary. The scheme reflects the suggestion of the International Classification for Seasonal Snow on the Ground (ICSSG) guide, in an effort to be consistent with the rest of the snow science community. Last, the colors, which represent differing grain habits, are also grouped numerically, and those associated sample numbers are listed on the figure. Thus, the figure is not reliant on the color scheme; a juxtaposition of Table 1 with Fig. 9 would allow the figure to be easily interpretable even for a reader with monochromacy.